

# Quantification of surface water volume changes in the Mackenzie Delta using satellite multi-mission data

Cassandra Normandin[1], Frédéric Frappart[2, 3], Bertrand Lubac[1], Simon Bélanger[4], Vincent Marieu[1], Fabien Blarel[3], Arthur Robinet[1] and Léa Guiastrennec-Faugas[1]

[1] EPOC, UMR 5805, Université de Bordeaux, Allée Geoffroy Saint-Hilaire, 33615 Pessac, France
[2] GET-GRGS, UMR 5563, CNRS/IRD/UPS, Observatoire Midi-Pyrénées, 31400 Toulouse, France
[3] LEGOS-GRGS, UMR 5566, CNRS/IRD/UPS, Observatoire Midi-Pyrénées, 31400 Toulouse, France
[4] Dép. Biologie, Chimie et Géographie, groupe BOREAS and Québec-Océan, Université du Québec à Rimouski, 300 allée des ursulines, Rimouski, Qc, G5L 3A1, Canada

*Correspondence to*: Cassandra Normandin (cassandra.normandin@u-bordeaux.fr)

**Abstract.** Quantification of surface water storage in extensive floodplains and their dynamics are crucial for a better understanding of global hydrological and biogeochemical cycles. In this study, we present estimates of both surface water extent and storage combining multi-missions remotely-sensed observations and their temporal evolution over more than 15 years in the Mackenzie Delta. The Mackenzie Delta is located in the North West of Canada and is the second largest delta in the Arctic Ocean. The delta is frozen from October to May and the recurrent ice break up provokes an increase of the river's flows. Thus, this phenomenon causes intensive floods along the delta every year with dramatic environmental impacts. In this study, the dynamics of surface water extent and volume analyzed from 2000 to 2015 by combining multi-satellite information from MODIS multispectral images at 500 m spatial resolution and river stages derived from ERS-2 (1995-2003), ENVISAT (2002-2010) and SARAL (since 2013) altimetry data. The surface water extent (permanent water and flooded area) peaked in June with an area of 9,600 km² (+/- 200 km²) on average, representing approximately 70% of the delta's total surface. Altimetry-based water levels exhibit annual amplitudes ranging from 4 m in the downstream part to more than 10 m in the upstream part of the Mackenzie Delta. A high overall correlation between the satellite-derived and in situ water heights (R>0.84) is found for the three altimetry missions. Finally, using altimetry-based water levels and MODIS-derived surface water extents, maps of interpolated water heights over the surface water extents are produced. Results indicate a high variability of the water height magnitude that can reach 10 meters compared to the lowest water height in the upstream part of the delta during the flood peak in June. Furthermore, the total surface water volume is estimated and shows an annual variation of approximately 8.5 km³ during the whole study period, with a maximum of 14.4 km³ observed in 2006. The good agreement between the total surface water volume retrievals and in situ river discharges (R=0.66) allows validating this innovative multi-mission approach and highlights the high potential to study the surface water extent dynamics.

**Key words:** Mackenzie Delta, surface water extent, multispectral imagery, satellite altimetry, surface water volume





## 1 Introduction

Monitoring the temporal and spatial variations in water stored or transiting, such as rivers, floodplain wetlands and lakes is essential (Seyler et al., 2008). Although wetlands and floodplain cover only 6% of the surface of the Earth (OECD, 1996), they are a key reservoir in hydrological and biogeochemical cycles. These areas play a major role in flood flow alteration, sediment stabilization settling, water quality, groundwater recharge and discharger (Maltby, 1990; Bullock and Acreman, 2003).

The Mackenzie Delta, a floodplain system located in Canada's western Arctic, is flat and composed of numerous lakes (Are and Reimnitz, 2000) and host large populations of birds, fishes and mammals (Beltaos, 2013; Squires et al., 2009). These lakes are ecologically sensitive environments largely controlled by river water (Squires et al., 2009). The Mackenzie Delta is ice covered during 7-8 months per year (Emmerton et al., 2007). The Mackenzie River flows northward from relatively warm area toward colder northern regions. The river ice break up starts in the South in March-April and progresses to the North to reach the delta sometime in May-June (Pavelsky and Smith, 2004). During the freshet, the freshwater meets an ice dam that was formed in winter along the shoreline of the Beaufort Sea known as the Stamukhi zone (Carmack and Macdonald, 2002) creating a large scale flooding of the delta. Ice breakup related floods can increase water levels to much higher elevations than open-water floods (Beltaos and Carter, 2009). This is one of the most important annual hydrologic events in cold regions (Muhammad et al., 2016). Floods are at the origin of significant surface water storage in the delta (Holmes et al., 2012; Pithan and Mauritsen, 2014).

These important recurrent floods replenish delta lakes with river water, sediments, and nutrients and play an important role in the maintenance of their ecosystems (Beltaos et al., 2012). It generates many environmental benefits such as geochemical land deposition, groundwater recharges, but also detriments such as infrastructure damages and lost economic activities (Kääb and Prowse, 2011).

Deltas are vulnerable to both anthropogenic and natural forcing such as socio-economic infrastructure development and global warming. In Arctic, the latter is particularly severe due to the polar amplification processes and complex positive feedback loops (Holmes et al., 2012). This system is undergoing important changes as the increase of precipitation at high latitudes, increase river discharge and melting of stock ices on land and sea (Stocker and Raible, 2005). These changes may induce an acceleration of the hydrologic cycle (Stocker and Raible, 2005). River discharge may increase from 18 to 70% from now to the end of the century (Peterson et al., 2002). Thus, the understanding of these dynamic environments is a societal and scientific stake to anticipate and manage their evolutions at medium and long term time scales.

Mapping surface water extent at the Mackenzie Delta scale is an important issue. However, it is nearly impossible with traditional methods in such a large and heterogeneous environment. Satellite remote sensing methods are the only way to solve this problem. Remote sensing has proven a strong potential to detect and monitor floods during the last two decades (Alsdorf et al., 2007; Smith, 1997). Typically, two kinds of sensor are used to map flooded area at high and moderate resolutions: passive multispectral imagery and active Synthetic Aperture Radar (SAR). The spectral signature of the surface reflectance is used to discriminate between water and land (Rees, 2001). The SAR images provide valuable information on the nature of the observed surface through the backscattering coefficient (Ulaby et al., 1981).

If space missions of radar altimetry were mainly dedicated to estimate ocean surface topography (Fu and Cazenave, 2001), it is now commonly used for monitoring inland waters levels (Birkett, 1995 ; Cazenave et al., 1997; Frappart





et al., 2006a; Santos da Silva et al., 2010; Crétaux et al., n.d.; Frappart et al., 2015b). Several studies have shown the possibility to measure water levels variations in lakes, rivers and flooding plains (Frappart et al., 2006b, 2015a; Santos da Silva et al., 2010). In the present paper, satellite multispectral imagery and altimetry are used in synergy to quantify the spatial extent of surface water extents and the surface water volumes of the Mackenzie Delta and

analyze their temporal variations. In the past, this approach has been successfully applied in tropical (e.g., the Amazon  (Frappart et al., 2012), Mekong  (Frappart et al., 2006b)) and peri-Arctic (e.g. the Lower Ob' basin, (Frappart et al., 2010) major river basins. But, the originality and novelty of the study is the use of multi-space mission data at high spatial and temporal resolutions over a long time series allowing to improve the robustness of retrievals.

**2 Study region**

The Mackenzie Delta is located in the northern part of Canada, extending between latitudes 67° and 70° N and longitudes 136° and 133° W (Figure 1a). It covers an area of 13,135 km² (Emmerton et al., 2008) and is the second biggest delta of Arctic with a length of 200 km and a width of 80 km, after the Lena Delta in Siberia (Emmerton et al., 2008). The average annual air temperature and precipitation are -8.8°C and 250 mm (50% as snow) in Inuvik,

Northwest Territories (http://wateroffice.ec.gc.ca/). The delta is a complex of multiple channels and numerous shallow and small lakes (over 49,000 lakes), covering nearly ~50% of the delta area (Emmerton et al., 2007). These lakes are continuously changing, through geomorphological processes of sediment deposits within lakes, division of large lakes into a number of smaller ones, enlarging of lakes due to melting of permafrost and creation of new lakes as channels are abandoned (Marsch, 1998).

This environment is also one of the most productive ecosystems in northern Canada with large populations of birds, fishes and mammals, which are critical resources for local population (Squires et al., 2009). The delta is dominated by permafrost and oil, and gas extraction activities are disturbing the environment (Kuenzer et al., 2015). The floodplain is composed of silt and sand, covered by boreal species of spruce (*Picea*), alder (*Alnus*), willow (*Salix*), birch (*Betula*), poplar (*Populus*), *Equisetum* and tundra species north of the tree line (Nguyen et

al., 2009).

The Mackenzie Delta is mainly drained by the Mackenzie River (90% of the delta's water supply) and Peel River (8% of the delta's water supply, Emmerton et al., 2007). The Mackenzie Delta channels have very mild slopes (-0.02 m/km, (Hill et al., 2001). The Mackenzie River ranks 4th in the Arctic with a freshwater discharge of 330 km$^3$.year$^{-1}$ and 18th in the world (Macdonald and Yu, 2006). The Mackenzie River begins in the Great Slave Lake

and then, flows through the North West territories before reaching the Beaufort Sea. It exports 130x10$^6$ tons.year$^{-1}$ of sediments to the Beaufort Self, which represents the main source of sediments in the Arctic (Macdonald and Yu, 2006).

The Mackenzie River has a strong seasonality in term of discharge due to snowmelt, from about 5,000 m$^3$.s$^{-1}$ in winter up to 40,000 m$^3$.s$^{-1}$ in June during the ice breakup for wet years (Figure 1b, Macdonald and Yu, 2006) ;

Goulding et al., 2009a ; Goulding et al., 2009b ; Beltaos et al., 2012). The Stamukhi is responsible for recurrent floods in the Mackenzie Delta. At the flood peak, 95% of the delta surface is likely to be covered with water (Macdonald and Yu, 2006). Water level peaks are mainly controlled by ice breakup effects and secondary by the amount of water contained in snowpack (Lesack and Marsh, 2010). Floods have significant physical, chemical and biological effects on Delta lakes (Marsch, 1998).





### 3 Data sets

### 3.1 Multispectral imagery

### 3.1.1 MODIS

The MODerate resolution Imaging Sensor (MODIS) is a spectroradiometer, part of the payload of the Aqua (since
2002) and Terra (since 1999) satellites. The MODIS sensor measures radiances in 36 spectral bands. In this study,
the MOD09A1 product (8-day binned level 3, version 6) derived from Terra and Aqua satellites raw radiance
measurements were downloaded from the United States Geological Survey (USGS) Earthexplorer website
([http://earthexplorer.usgs.gov](http://earthexplorer.usgs.gov)). It consists in gridded, atmospherically corrected surface reflectance acquired in 7
bands from visible to short wave infrared (2155 nm) at a 500 m spatial resolution. This product is obtained
combining for each wavelength the best surface reflectance data of every pixel acquired during an 8-day period.
Each MODIS tile covers an area of 1200 km by 1200 km. Two tiles (h12v02 and h13v02) are used to cover the
whole study area. In this study, 223 composites, acquired during the ice-free period from June to September over
the 2000-2015 time-span, are used.

### 3.1.2 OLI

The Landsat-8 satellite is composed of two Earth-observing sensors, the Operational Land Imager (OLI) and
Thermal InfraRed Sensor (TIRS). This satellite was launched in February 2013 and orbits at an altitude of 705 km.
The swath is 185 km and the whole Earth surface is covered every 16 days.
The OLI/TIRS sensors measure in 11 spectral bands in the visible (450-680 nm), near infrared (845-885 nm) and
short wave infrared portions (1,560-2,300 nm) of the electromagnetic spectrum. In this study, the Landsat 8/OLI
surface reflectance products were downloaded from the Landsat-8 USGS portal (http://earthexplorer.usgs.gov/).
The multispectral spatial resolution is 30 m and 15 m for panchromatic band. Two images are necessary to cover
the Mackenzie Delta.
Landsat-8 mission is characterized by a lower revisit time than Terra and Aqua mission. Thus, associated with a
high occurrence of clouds over the study area, Landsat-8 yields to a small amount of high-quality data. OLI images
cannot be consequently used in this study to monitor land water surface temporal changes. In this context, MODIS
represent a relevant alternative to OLI despite a lower spatial resolution. However, available high-quality OLI data
have been used to compare and validate MODIS land water surfaces.

### 3.2 Radar altimetry data

### 3.2.1 ERS-2

The ERS-2 satellite (European Remote Sensing) was launched in 1995 by the European Space Agency (ESA). Its
payload is composed of several sensors, including a radar altimeter (RA), operating at Ku-band (13.8 GHz). It was
orbiting sun-synchronously at an altitude of 790 km with an inclination of 98.54° with a 35-day repeat cycle. This
orbit was ERS-1's orbit with a ground-track spacing about 85 km at the equator. ERS-2 provides observations of
the topography of the Earth from 82.4° latitude north to 82.4° latitude south. ERS-2 data are disposable from 17
May 1995 to 9 August 2010 but after 22 June 2003, the coverage is limited.





### 3.2.2 ENVISAT

Envisat mission was launched on March 1st 2002 by ESA. This satellite carried 10 different instruments including the advanced radar altimeter (RA-2). It was based on the heritage of ERS-1 and 2 satellites. RA-2 was a nadir-looking pulse-limited radar altimeter operating at two frequencies at Ku- (13.575 GHz) and S-(3.2 GHz) bands.

Its goal was to collect radar altimetry over ocean, land and ice caps (Zelli, 1999). Envisat remained on its nominal orbit until October 2010 but RA-2 stopped operating correctly at S-band in January 2008. Its initial orbital characteristics are the same as for ERS-2.

### 3.2.3 SARAL

SARAL mission was launched on 25 February 2013 by a partnership between CNES (Centre National d'Etudes

Spatiales) and ISRO (Indian Space Research Organization). Its payload comprised the AltiKa radar altimeter and bi-frequency radiometer, and a triple system for precise orbit determination: the real-time tracking system DIODE of DORIS instrument, a Laser Retroflector Array (LRA), and the Advance Research and Global Observation Satellite (ARGOS-3). AltiKa is the first radar altimeter to operate Ka-band (35.75 GHz). It is a solid-state mono-frequency altimeter that provides precise range estimates (Verron et al., 2015). SARAL orbit was earlier utilized

by ERS-1 & 2 and ENVSAT missions with a track spacing of 85 km at the equator (Verron et al., 2015). It has been put on a drifting orbit since 4 July 2016.

Altimetry data used here are contained in the Geophysical Data Records (GDRs) and are the followings:

- cycle 001 (17/05/1995) to cycle 085 (07/08/2003) for ERS-2 from the reprocessing of the ERS-2 mission raw waveform performed at Centre de Topographie de l'Océan et de l'Hydrosphère (CTOH) (Frappart et al.,

20 2016)

- GDR v2.1 for ENVISAT from cycle 006 (14/05/2002) to cycle 094 (21/10/2010)

- GDR E for SARAL from cycle 001 (15/03/2015) to cycle 027 (14/10/2015)

These data were made available by CTOH (http://ctoh.legos.obs-mip.fr/). Data were acquired along the altimeter track at 18, 20 and 40 Hz for ENVISAT, ERS -2 and SARAL respectively (high-frequency mode commonly used

over land and coastal areas where the surface properties are changing more rapidly than over the open ocean). They consist of the satellite locations and acquisition times and all the parameters necessary to compute the altimeter heights (see Section 4.3).

### 3.3 In situ water levels and discharges

The altimetry-based water level time-series derived from radar altimetry were compared to gauge record from in

situ stations for validation purpose. Data from 10 gauge stations were found in a close vicinity of altimetry virtual stations (at a distance lower than 20 km along the streams). Virtual stations are built at intersections between an orbit groundrack and a water body (lake, river and floodplain) (Crétaux et al., in press). Besides, surface water storage variations were compared to the rivers flow entering the delta summing the records from 3 gauge stations located in upstream part of the delta. Daily data of water level and discharge were downloaded for free from the

Canadian government website (http://wateroffice.ec.gc.ca).





## 4 Methods

### 4.1 Quantification of surface water extent

Multispectral imagery is commonly used for delineating flood extent using spectral indices (e.g., Frappart et al., 2006b; Sakamoto et al., 2007; Crétaux et al., 2011; Pekel et al., 2014; Verpoorter et al., 2014) ; Ogilvie et al.,
2015; Pekel et al., 2014). In this study, we used the approach proposed by (Sakamoto et al., 2007) to monitor the land water surface extent in the Mackenzie Delta (Figure 2). This approach is based on the application of thresholds on the Enhanced Vegetation Index (EVI), the Land Surface Water Index (LSWI) and the Difference Value between EVI and LSWI (DVEL=EVI-LSWI) to determine the status (non-flooded, mixture, flooded and permanent water body) of any pixel in an 8-day MODIS composite image of surface reflectance. As the spectral response of the
Near Infra-Red (NIR) and short-wave infrared (SWIR) bands are highly dependent on the Earth surface nature, in particular water versus soil/vegetation surfaces, their complementary was used to define LSWI. For instance, the surface reflectance presents low values (a few percentages) over non-turbid water bodies and high values (a few tens of percentage) over vegetation feature in the NIR spectral bands. The spectral response in the SWIR is mainly dominated by strong water absorption bands, which is directly sensitive to moisture content in the soil and the
vegetation. For water surface, the signal in the SWIR is assumed to be zero even in turbid waters (Wang and Shi, 2005). Thus, LSWI is expected to get values close to 1 for water surfaces and lower values for non-water surfaces. The two indices, used in this approach, are defined as follows (Huete et al., 1997; Xiao et al., 2005):

$$\text{EVI} = a \times \frac{\rho_{NIR} - \rho_{red}}{\rho_{NIR} + b \times \rho_{red} - c \times \rho_{blue} + d} \tag{1}$$

$$\text{LSWI} = \frac{\rho_{NIR} - \rho_{SWIR}}{\rho_{NIR} + \rho_{SWIR}} \tag{2}$$

Where for MODIS, $\rho_{blue}$ is the surface reflectance value in the blue (459-479 nm, band 3), $\rho_{red}$ is the surface reflectance value in the red (621-670 nm, band 1), $\rho_{NIR}$ is the surface reflectance value in the NIR (841-875 nm, band 2), and $\rho_{SWIR}$ is the surface reflectance in the SWIR (1628-1652 nm, band 6). For OLI, $\rho_{blue}$, $\rho_{red}$, $\rho_{NIR}$ and $\rho_{SWIR}$ are associated to channel 2 (452-512 nm), channel 4 (636-673 nm), channel 5 (851-879 nm), and channel 6 (1570-1650 nm). *a, b, c* and *d* constants equal to 2.5, 6, 7.5 and 1, respectively, for both MODIS and OLI (USGS,
product guide).

To process multispectral images, the first step consists in removing the cloud-contaminated pixels applying a cloud masking based on a threshold of the surface reflectance in the blue band ($\rho_{blue} \geq 0.2$). Then, spectral indices are computed. Note that, contrary to (Sakamoto et al., 2007), no smoothing was applied on spectral indices time-series. In a second step, the identification of the status of each pixel is performed applying thresholds on EVI, LSWI and
their differences (Figure 2), which reduce the noise component. Thresholds determined by Sakamoto et al. (2007) were validated for our study site using the MODIS and OLI images acquired on 04/07/2013 and 01/07/2013 respectively (Figure 3). Histograms show a similar bi-modal distribution for both EVI, LSWI and EVI-LSWI between MODIS and OLI 500 m (Figure 3 and 4). For EVI, pixels with a value lower than 0.1 are clearly associated with water land surfaces, while pixels with a value higher than 0.3 are associated with soil and vegetation features.
Other pixels, with an EVI value comprised between 0.1 and 0.3, are identified as mixed surface types. For LSWI, pixels with a value higher than 0.5 are clearly associated with water land surfaces, while pixels with a value lower than 0.3 are associated with vegetation features or soil land surfaces when LSWI values are negative. Other pixels, with an LSWI value comprised between 0.3 and 0.5, are identified as mixed surface types. Contrary to what was found by Sakamoto et al. (2007) in the Mekong Basin, no negative value of LSWI were observed over our study





area. This threshold was not applied in this study. For EVI-LSWI, pixels with a value lower than -0.05 are represented water land surface and values comprised between -0.05 and 0.1 are associated to mixture pixels. Other pixels, with values higher than 0.1 are represented vegetation features or soil land surfaces (Figure 4). Each pixel was then classified in two main categories: non-flooded (EVI > 0.3 or EVI ≤ 0.3 but EVI – LSWI > 0.05) and

water-influenced (EVI ≤ 0.3 and EVI – LSWI ≤ 0.05 or EVI ≤ 0.05) (Figure 2). The second category was divided into three sub-classes: mixed pixels (0.1< EVI ≤ 0.3), flooded pixels (EVI ≤ 0.1) and permanent water bodies (e.g. lake, river and sea), when the total duration of a pixel classified as flooded is longer than 70 days out of 105 day for the study period. This total duration correspond roughly to 2/3 of the study period, as proposed by Sakamoto et al. (2007). The spatio-temporal variations of floods have been characterized for the months included between

June and September over the 2000-2015 period.

Thereafter, we define in this paper land water surface as permanent water bodies with flooded areas although inundated surfaces including only inundated areas.

### 4.2 Validation of MODIS retrievals using OLI

Evaluation of the performance of the land water surface detection from MODIS is based on the comparison

between land surface water estimated from MODIS at a 500 m-resolution, OLI at a 30 m-resolution, and OLI re-sampled at 500 m-resolution. For the validation purpose, MODIS and OLI images are selected when (1) the time-difference between the acquisitions of two satellite images is lower than 3 days and (2) the presence of cloud over the area is lower than 5%. Following these criteria, only two cloud-free OLI composites were selected between 1st July 2013 and 2nd August 2013.

### 4.3 Satellite-derived water level time-series in the Mackenzie Delta

The concept of radar altimetry is explained below. The radar emits an electromagnetic (EM) wave towards the surface and measures the round-trip time (Δt) of the EM wave. Taking into account propagation corrections caused by delays due to the interactions of electromagnetic wave in the atmosphere, and geophysical corrections, the height of the reflecting surface (h) with reference to an ellipsoid can be estimated as (Crétaux et al., submitted):

$$h = H - (R + \sum \Delta R_{propagation} + \Delta R_{geophysical}) \qquad (3)$$

where H is the satellite center of mass height above the ellipsoid, R is the nadir altimeter range from the satellite center of mass to the the surface taking into account instrument corrections (R=$c\Delta t$/2 where $c$ is the light velocity in the vacuum), $\sum \Delta R_{propagation}$ is the sum of the geophysical and environmental corrections applied to the range, respectively.

$$\sum \Delta R_{propagation} = \Delta R_{ion} + \Delta R_{dry} + \Delta R_{wet} \qquad (4)$$

where $\Delta R_{ion}$ is the atmospheric refraction range delay due to the free electron content associated with the dielectric properties of the ionosphere, $\Delta R_{dry}$ is the atmospheric refraction range delay due to the dry gas component of the troposphere, $\Delta R_{wet}$ is the atmospheric refraction range delay due to the water vapor and the cloud liquid water content of the troposphere.

$$\sum \Delta R_{geophysical} = \Delta R_{solid\ Earth} + \Delta R_{pole} \qquad (5)$$

Where $\Delta R_{solid\ Earth}$ and $\Delta R$pole are the corrections respectively accounting for crustal vertical motions due to the solid Earth and pole tides. The propagation corrections applied to the range are derived from model outputs: the Global Ionospheric Maps (GIM) and Era Interim from the European Centre Medium-Range Weather Forecasts



(ECMWF) for the ionosphere and the dry and wet troposphere range delays respectively. The changes of the altimeter height h over the hydrological cycles are related to variations in water level. Here, the Multi-mission altimetry Processing Software (MAPS) was used to precisely select valid altimetry data at every virtual station locations (see part 3.3) series in the Mackenzie Delta (Frappart et al., 2015b). Data processing consists in four steps:

- The rough delineation of the river/lake cross-sections with overlaying altimeter tracks using Google Earth,
- The loading of the altimetry over the study area and the computation of the altimeter heights from the raw data contained in the GDRs,
- Valid altimetry data were selected through a refined process. Outliers and measurements over non-water surfaces are delete, based on visual inspection.
- The computation of the time-series of water level.

**4.4 Surface water volume storage**

The approach used to estimate the anomalies of surface water volume is based on the combination of the surface water extent derived from MODIS images with altimetry-based water levels estimated at virtual stations distributed all over the delta (Figure 6). Surface water level maps were computed from the interpolation of water levels over the land water surfaces using an inverse-distance weighting spatial interpolation technique following (Frappart et al., 2012). Hence, water level maps were produced every 8 days from 2000 to 2015. For each water pixel, the minimal height of water during 2000-2015 is estimated. As ERS-2, ENVISAT and SARAL had a repeat cycle of 35 days, water levels are linearly interpolated every 8 days to be combined with the MODIS composite images. Surface water volume time series are estimated over the Mackenzie Delta following (Frappart et al., 2012):

$$V = \sum_{j \, \epsilon \, S} [h \, (\lambda_j, \varphi_j) \, - \, h_{min} \, (\lambda_j, \varphi_j)]. \, \delta_j \, . \, \Delta S \qquad (6)$$

Where V is the anomaly of surface water volume (km3), S is the surface of the Mackenzie Delta (km²), $h \, (\lambda j, \varphi j)$ the water level, $h_{min} \, (\lambda_j, \varphi_j)$ the minimal water level for the pixel of coordinates $(\lambda_j, \varphi_j)$ inside the Mackenzie Delta, $\delta_j$ equals 1 if the $j^{th}$ pixel is associated to permanent water body/inundated and 0 if not and $\Delta S$ the pixel surface (0.25 km²).

**5. Results and discussion**

**5.1 MODIS-based land water extent**

Map of annual average of land water surface, composed of inundated and permanent water bodies, was obtained at spatial and temporal resolutions of 500 m and 8 days respectively from June to September over the 2000-2015 period (Figure 5a). Map of annual average of land water surface duration along with associated standard deviation over 2000-2015 during ice-free period of three months and half (105 days) is presented in Figure 5b. Permanent water bodies (i.e., identified as land water surface more than 70 days annually) are located along the Mackenzie River main channel, its tributaries (Reindeer, Peel, Middle and East Channels) and major lakes of the Delta. The longer water areas (i.e., identified as flooded between 30 and 70 days annually) are surrounding permanent water bodies. Other areas of the delta are annually inundated up to 30 days (Figure 5a). The map of standard deviation of the annual flood duration shows ranges from a few days over the areas affected by floods during a short time span to 15 days close to permanent water bodies (Figure 5b).





Maps of errors made on land water surface duration with associated standard deviation are shown in Figure 5c and 5d over 2000-2015. Mixture pixels have been used to calculate for each pixel the error on land water surface duration. Standard deviation of error is presented in Figure 5d. Maximal error and error standard deviation is obtained for pixels of potential flooding area in the delta. If short differences – lower than $20 \pm 12$ days – can be

observed in the downstream part of the delta (over 69°N), longer differences (30 to $50 \pm 15$ to 20 days) are present in the upstream part. They can be attributed to the presence of small permanent lakes in this area (see the discussion on the validation of the surface water extent (see sub –section 5.2).

Maps of difference between the duration of extreme land water surface and the mean land water surface duration from 2000 to 2015 were estimated for the large historic flood that occurred in 2006 (Figure 5e) and for the minimal

flood that occurred in 2010 (Figure 5f). The whole Mackenzie Delta was practically covered of water in 2006, whereas large areas, especially in the downstream part of the delta, were not inundated in 2010 (Figure 5f).

Time series of land water surfaces of the Mackenzie Delta were derived from the 8-day maps of land water surface extent (Figure 6). Each year, water surface extent is maximum in June in response to the snow melt that occurred in May (between day of year, DOY, 110 and 130 on average) in the Delta and decreases to reach a minimum in

September, as previously observed by Goulding et al., 2009b, Goulding et al., 2009a. Land water surface extent varies from 1,500 to 14,284 km² between 2000 and 2015 along the hydrological cycle (Figure 6). On average, during the study period, maximum surface water extent is ~9,600 km². The largest water surface extent was reached in June 2006 with an inundated area of 14,284 km², which represents ~85% of the delta total surface (Figure 6). Large surface water extents (~12,500 km²) were also detected in 2011 and 2013 in accordance with high discharge

peaks reported these years (http://wateroffice.ec.gc.ca/) and the historic inundation that occurred in Aklavik in 2006 (Beltaos and Carter, 2009).

### 5.2 Validation of land water surface

Surface water extent (the sum of permanent bodies and inundated areas) were also estimated applying the approach described in sub-section 3.1 for OLI images at 30 m of spatial resolution, and resampled at 500 m of spatial

resolution. They were compared to MODIS-based surface water extent for the closest date (Table 1). Figure 7a, 7b and 7c present the maps of the surface water extent determined using MODIS, OLI 500 m and OLI 30 m respectively, acquired in July 2013. Medium and large scale (with a minimal size of 300 m) land water features are well detected as displayed on the zoomed part of images. Figure 7c present a zoom of surface water extent using OLI 30 m with permanent and inundated bodies. Surface water extent from OLI 500 m and MODIS are

similar for both dates with differences lower than 20% (Table 1). For example in July 2013, land water surface is about 4,499 km² for OLI 500 and 3,798 km² for MODIS (Table 1). Percentages of common detection of surface water were estimated for the pixels detected as land water surface in the pair of satellite images. These percentages are 73 and 74 % for July 2013 and August 2013, respectively. These results highlight the robustness of the method of Sakamoto et al. (2007) for accurate land water surface retrievals. These surface water extent have been

compared with surface water extent (channels and wetlands) determined by Emmerton et al. (2007) in Table 1. For MODIS, differences are lower than 15% and for OLI 500, differences are about 25% (Table 1).

However, the comparison between surface water extent estimated from OLI 30 m and MODIS 500 m shows important differences. In July 2013, surface water extent is about 3,798 km² from MODIS and 7,685 km² from OLI 30. The surface extents are higher for OLI 30 by a factor of 2 (Table 1). According to Emmerton et al., 2007,





the Mackenzie Delta is composed of 49,000 lakes with a mean area of 0.0068 km² and 40% of the total number of lakes have an area inferior to 0.25 km². The pixel sizes of OLI 30 m and MODIS 500 m are approximately 0.0009 km² and 0.25 km², respectively. Thus, the high difference between the land water surfaces detected using OLI 30 m and MODIS is probably associated to a spatial sample bias. Small-scale water features detected from OLI cannot be detected from MODIS due to a lower spatial resolution. Surface water extent determined using OLI 30 have been compared to Emmerton et al. (2007) surface water extent (including channels, wetlands and lakes) and differences lower than 15 % are found (Table 1).

In order to investigate the assumption of spatial sample bias associated with MODIS 500 m, a satellite validation of surface water extent is performed (Table 2). Permanent water and inundated surfaces have been calculated for MODIS, OLI 500 and OLI 30. For OLI 30 and OLI 500, pixels identified as surface water for the two dates are considered as permanent waters (Table 2). In July 2013, inundated surfaces are nearly equal, about 577 km² for MODIS, 690 km² for OLI 500 and 627 km² for OLI 30 (Table 2). In August, inundated surfaces are equal to 250 km² and are 2.5 more important than OLI 30 (98 km²), if we consider OLI 30 as truth.

### 5.3 Alimetry-based water levels

The Mackenzie Delta is densely covered with altimetry tracks from the ERS-2, ENVISAT and SARAL missions that all were on the same nominal orbit. Twenty-two, twenty-seven and twenty-four altimetry virtual stations were built at the cross-section of an altimetry track with a water body for these three missions respectively (see Figure 8 for their locations). A water level temporal series is obtained for each virtual station.

The quality of altimetry-based water levels was evaluated using *in situ* gauge records. Only six virtual stations are located near *in situ* stations (with a distance lower than 20 km) for ERS-2 data, ten for ENVISAT and eight for SARAL data. Characteristics of these virtual stations are given in Tables 3, 4 and 5 for ERS-2, ENVISAT and SARAL, respectively.

Altimetry-based water levels were validated using these virtual stations close enough (< 20 km) to *in situ* stations (6 comparisons for ERS-2, 10 for ENVISAT and 8 for SARAL). For ERS-2 and SARAL comparisons, the correlation r is low at the station 0114-c, i.e. -0.38 and 0.15 respectively.

For ERS-2, quite high correlation coefficients are obtained for 4 virtual stations out of 6, with r ≥ 0.69 and RMS ≤ 1 m (Table 3). For the two other stations, no correlation is observed (-0.38 and 0.08 for ERS-2-0114c and ERS-2-0200-d respectively with a RMS ≥ 1 m) (Table 3).

For ENVISAT, 8 out of 10 stations have a correlation coefficient ranging between 0.66 and 0.93 (Table 4). Except for ENV-0572-a, which is located 22 km away from the nearest in situ station, higher correlations were found when the river is larger at the VS (Table 4). For example, ENV-0114-b exhibits a negative correlation (r = -0.27) where the cross-section was only 150 m width (Table 4). This station is also located near the city of Inuvik. The presence of the town in the altimeter footprint could exert a strong impact on the radar echo and explain this low correlation.

For SARAL, 5 out of 6 virtual stations have a good correlation r coefficient higher than 0.76 with a low RMS (Table 5) due to its narrower footprint with an increase of the along-track sampling.

Comparisons between water levels derived from altimetry and *in-situ* are shown for two stations for ERS-2 (called ERS-2-0744-a and ERS-2-0439-a; Figure 9a and 10a), three for ENVISAT (ENV-0744-a, ENV-0439-a and ENV-0028-a; in Figure 9b, 10b and 11) and two for SARAL (SARAL-0744-a and SARAL-0439-a; Figure 9c and 10c).





Virtual station 0744-a is located in the downstream part of the delta, 0439-a in the center and 0028-a in the upstream part (Figure 8). For each station, water levels obtained by altimetry and water levels of *in situ* gauge are superposed (Figures 9, 10 and 11). Then, water level anomalies, which are computed as the average water level minus the water level, have been calculated for altimetry and in situ data.

The virtual station 0744-a is located in the North of the Mackenzie Delta (Figure 8). Water level time-series have been processed between 1995 and 2015 and compared to *in situ* data of the station 10MC010 for each mission ERS-2, ENVISAT and SARAL (Figure 9). *In situ* data are not continuous since river is frozen from October to April. With regard to altimetry, data have been acquired all the year but during frozen periods, water levels are unrealistic due to the presence of river ice. Thus, the processing is done only from the beginning of June to the end

of September as for multispectral imagery treatment. The correlation r between altimetry water levels and in situ levels is 0.88 for ERS-2, 0.93 for ENVISAT and 0.99 for SARAL (Tables 3, 4 and 5). For the three missions, RMS is weak, lower than 0.15 m (Tables 3, 4 and 5). At this station, the variation of water level is about 2 m on average with an important water level in June that decreases to September (Figure 10a, 10c and 10e).

The virtual station ERS-2-0439-a is in the center of the Mackenzie Delta and water levels time-series have been

done between 1995 and 2015 and compared to *in situ* data of the station 10MC008 for the three missions ERS-2, ENVISAT and SARAL (Figure 10). The correlation between altimetry water levels and water levels from in situ gauge is about 0.76 for ERS-2, 0.89 for ENVISAT and 0.96 for SARAL (Tables 3, 4 and 5). RMS is included between 0.35 and 0.5 m for the three missions. On average at this station, water levels variations are about 4 meters with a maximal water level in June that decreases to reach a minimal value in September (Figure 10a, 10c and

10e).

Water levels time-series between 2002 and 2010 at the virtual station ENV-0028-a located upstream of the Mackenzie Delta have been compared to *in situ* data of the station 10LC014 (Figure 11). A good correlation was found for this station too, with a coefficient correlation r of 0.83 and a RMS of 1.84 m (Table 4). For this station, variations of water levels are much higher with 9 m on average but reaching 12 m during the 2006 extreme event

(Figure 11a). Water levels time-series have been done only for ENVISAT mission since for the two others (ERS-2 and SARAL), altimetry water levels were not consistent with values around 70 meters. Therefore, water levels determined by altimetry and water level from *in situ* gauge have a difference, probably explained by the distance between virtual station and *in situ* gauge (16.31 km) since the slope is about -0.02m/km in the Delta (Hill et al., 2001). Moreover, the seasonal cyclic thawing and freezing of the active layer causes cyclic settlement and heave

at decimeters levels, estimated to 20 cm (Szostak-Chrzanowski, 2013).

To summarize, water levels time-series were presented for three stations along the Mackenzie Delta (Figure 9, 10 and 11). For all stations and RA missions, a strong seasonal cycle can be seen, with a maximum water level reach in June after the snow-melt that decreases to reach a minimal value in September, in good accordance with the hydrological cycle of the Mackenzie Delta. The Delta is frozen from October to May and during spring-early

summer, the freshwater meets an ice dam that was formed in winter, what provokes river discharges variations from 5,000 $m^3$ to 25,000 $m^3$ on average (http://wateroffice.ec.gc.ca/). Then, these important variations provoke water levels increase and important floods each year in the delta. However, water levels variations as revealed from RA are not equal over the delta. In the upstream part, variations are 9 m on average, 4 meters in the center and 3 meters in the downstream part of the Mackenzie Delta.





### 5.4 Time series of surface water storage anomalies in the Mackenzie Delta

The minimum water level of each inundated pixel was determined over the observation period. 8-day surface water levels maps were created after subtracting the minimum water level to water level at time t, using MODIS-based flood extent and altimetry-derived water levels in the entire delta from June to September. Example of water level

maps are presented for 2006 at 4 different dates (in June, July, August and September), characterized as an historic flood (Figure 12).

Over the study period, water level maps show a realistic spatial pattern with a gradient of water level from south to north consistent with flow direction in the delta. On Figure 12a, in June 2006 for example, water levels are higher (about 5 m) upstream than downstream (about 0.5 m). The surface water storage reaches its maximal extent

in June (Figure 12a) and then decreases during the following months, reaching 1 m in September in the entire delta (Figure 12b, 12c and 12d).

The time series of surface water volume variations was estimated from 2000 to 2010 and then from 2013 to 2015, between June and September, following a similar approach as in Frappart et al., 2012 (Figure 13). Surface water storage was estimated from 2000 to 2003 using ERS-2 data, from 2003 to 2010 using ENVISAT data and from

2013 to 2015 using SARAL. Between 2010 and 2013, surface water storage could not be estimated due to lack of RA data over the delta. The time series of surface water volumes is presented in Figure 13. We assess the impact of the presence of a virtual station located in the upstream part of the delta and the inclusion of ERS-2 data on our satellite-based surface water volume estimation. For ERS-2 and SARAL data, no virtual station was created in the upstream part due to unreliable water levels in the. During the SARAL observation period, *in situ* water levels

from 10LC014 station were used. One curve corresponds to surface water volume with virtual station in the upstream part of the delta (2002-2015; red) and another one without virtual station in the upstream part of the delta (2000-2015; green). Correlations between river discharges and surface water volumes with and without (2002-2015) upstream virtual station are the same (0.66). Of the presence of a virtual station in the upstream part of the Mackenzie decreases the water volume by ~0.3 km³ on average (Figure 13). The correlation is lower (0.63) when

ERS-2 data are included in the analysis (2000-2015). The integration of ERS-2 data have a lower accuracy slight decrease the correlation between water storage and flux.

In term of temporal variability, a clear seasonal cycle is visible with a yearly maximum of water surface volume occurring in June (about 9.7 km³ on average), followed by a decrease until September (Figure 13). The peak generally corresponds to the presence of the extensive flood covering the delta in June, and during summer, the

volume decreases to reach its minimal in September (~0.2 km³). The largest surface water volume happened in 2006 with a volume of 14.4 km³ (Figure 13), known as an historic flood (Beltaos and Carter, 2009). These results showed that the satellite-based surface water volumes estimation are consistent with the Mackenzie River discharge, which is the main driver of the delta flooding.

### 5.5 Validation

Our results were compared to the ones estimated by Emmerton et al. (2007) under the assumption of a storage change as a rectangular water layer added to the average low-water volume for a stage variations from 1.231 m above sea level during low water period and 5.636 m above sea level at peak flood. Using this approach, Emmerton et al. (2007) found an increase in water volume of 14.14 km³ over the floodplains and 7.68 km³ over the channels. With our method, maximal water volume is around 9.6 km³ in average and can reach 14 km³. As it can be seen in



Figure11, water levels present a strong gradient over the delta and are, in average, lower than 5.636 m from Emmerton et al. (2007). The difference of approaches is likely to account for such discrepancy.

## 6. Conclusion

This study provides surface water estimates (permanent water of large features and inundated surfaces) dynamics both in extent and storage in the Mackenzie Delta from 2000 to 2015 using MODIS images at 500 m of spatial resolution and altimetry-based water levels. Surface water exhibits a maximal extent in the beginning of June and decreases to reach a minimal value in September. In June, the extent of land water surface is on average about 9,600 km². The highest value was observed in 2006 (~14,284 km²), during the historic flood described by (Beltaos and Carter, 2009). Despite the lower resolution of MODIS images in comparison with Landsat-8 ones, surface water extent estimates are quite similar using both sensors over the river channels and the floodplains with an underestimation of 20% is found for MODIS. But, the large number of small lakes is not detected using MODIS Virtual stations, or river/lake cross-section have been created across the Mackenzie Delta for the three radar altimetry missions (ERS-2, 1993-2003; ENVISAT, 2002-2010; SARAL, since 2013). The water levels determined by altimetry at those stations have been validated with *in situ* river levels with good correlation coefficient (> 0.8) for the three missions. The combination between land water extent determined by MODIS imagery and the water levels derived from altimetry has permitted to calculate the surface water volume in the Mackenzie Delta at 8-day temporal resolution. Temporal variations in surface water volume calculated from 2000 to 2015 showed a maximal volume in June (on average 9.6 km³) and a minimal volume in September (about 0.1 km³). A relatively strong correlation was found between surface water volume and the Mackenzie River discharges (R=0.66), suggesting Frthat the latter is the main driver of the delta flooding. Overall these results indicate that the satellite-based water volume estimation are consistent and can be used to monitor the recurrent flooding of large Arctic deltas.

The recent launches of Sentinel-1, Sentinel-2 and 3 offer new opportunities for flood monitoring at higher spatial (~10 m) and temporal (a few days) resolutions. Besides, these products provide a unique long-term dataset that allows a continuous monitoring of the changes affecting the surface water reservoir the launch of the NASA-CNES Surface Water and Ocean Topography (SWOT) mission in 2021. Associated with Aquatic color radiometry (Mouw et al., 2015), the approach developed here should provide useful information for the study of fluvial particle transport along the river-to-coastal ocean continuum and its potential impacts on ecosystems.

## 7. Acknowledgments

This study was supported by an internship grant from LabEX Côte (Université de Bordeaux) and a PhD grant from Ministère de l'Enseignement Supérieur et de la Recherche and also by the CNES TOSCA CTOH and the CNES OSTST FOAM grants. The authors also thank David Doxaran for fruitful discussion.

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




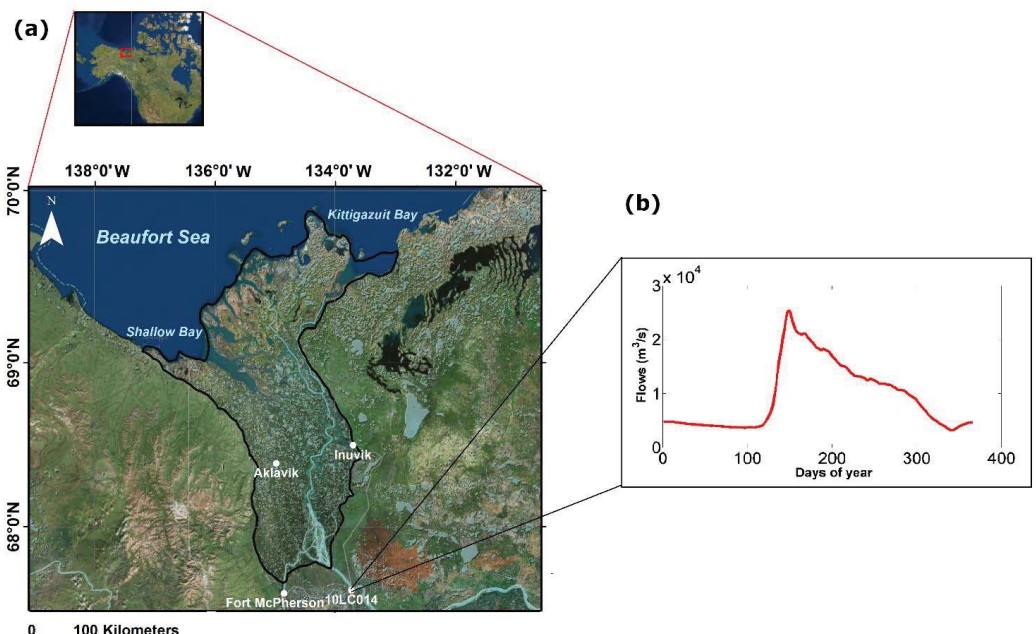

**Figure 1: (a) Location of the Mackenzie Delta at the mouth of the Mackenzie River in the Northwest Territories of Canada (b) Average annual river flow of the Mackenzie River at 10LC014 station (133°W, 67°N), 30 km upstream the Mackenzie Delta.**





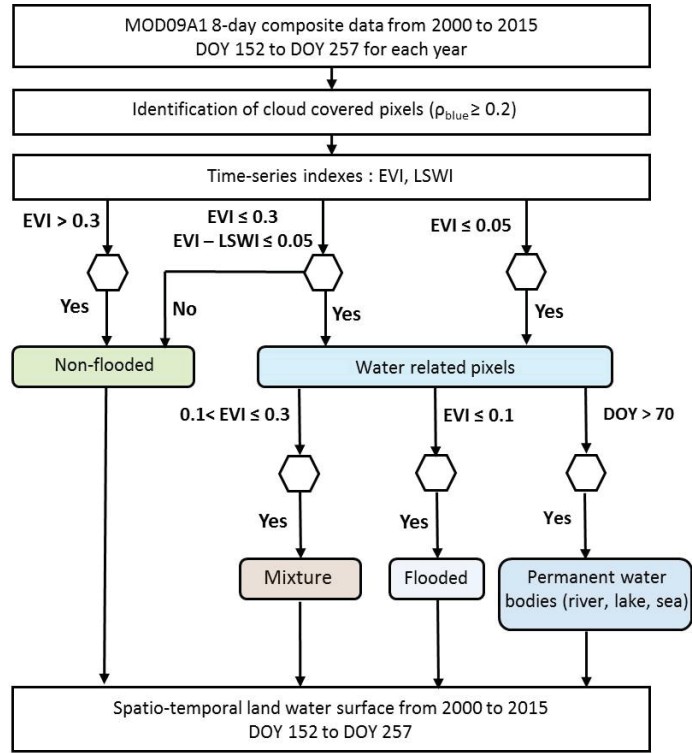

**Figure 2: Flow-chart of the method (adapted from Sakamoto et al., 2007) used to classify each pixel of the multispectral images acquired over the Mackenzie Delta in 4 categories (non-flooded, mixture, flooded and permanent water bodies) for each year from 2000 to 2015 using MODIS 8-day composite data from the day of the year (DOY) 169 to 257.**


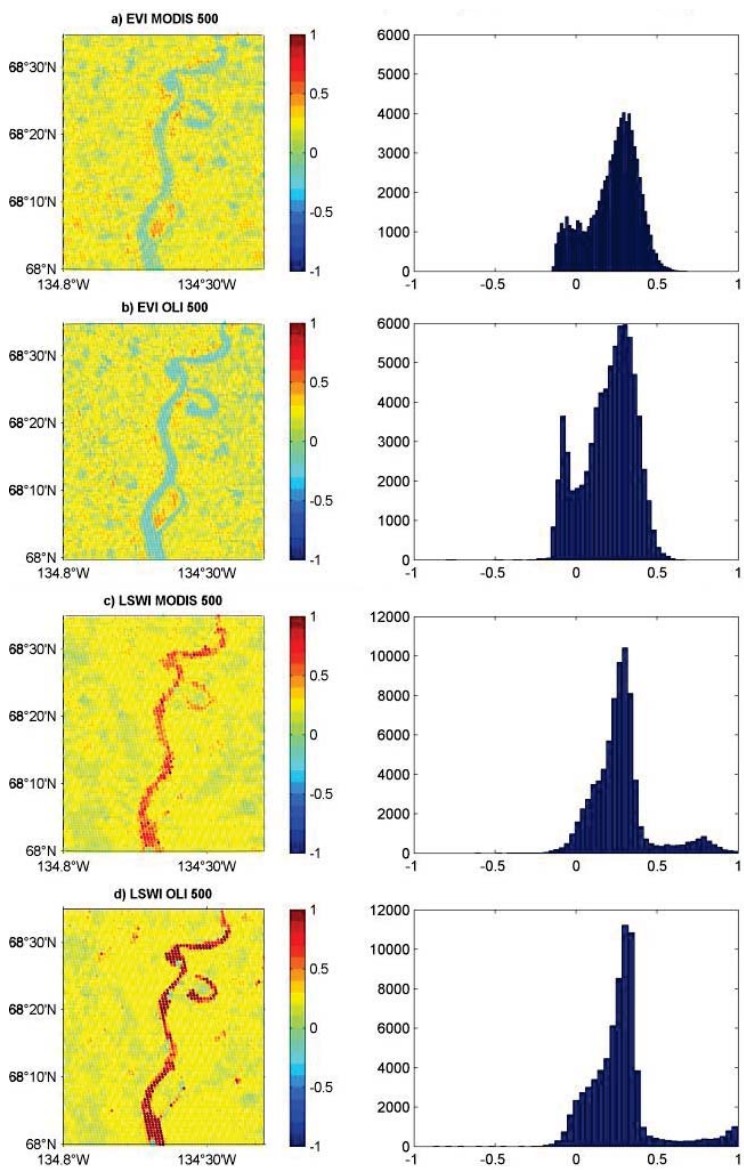

**Figure 3: Index map and associated histogram in July 2013 for (a) EVI for MODIS, (b) EVI for OLI 500, (c) LSWI for MODIS and (d) LSWI for OLI 500**



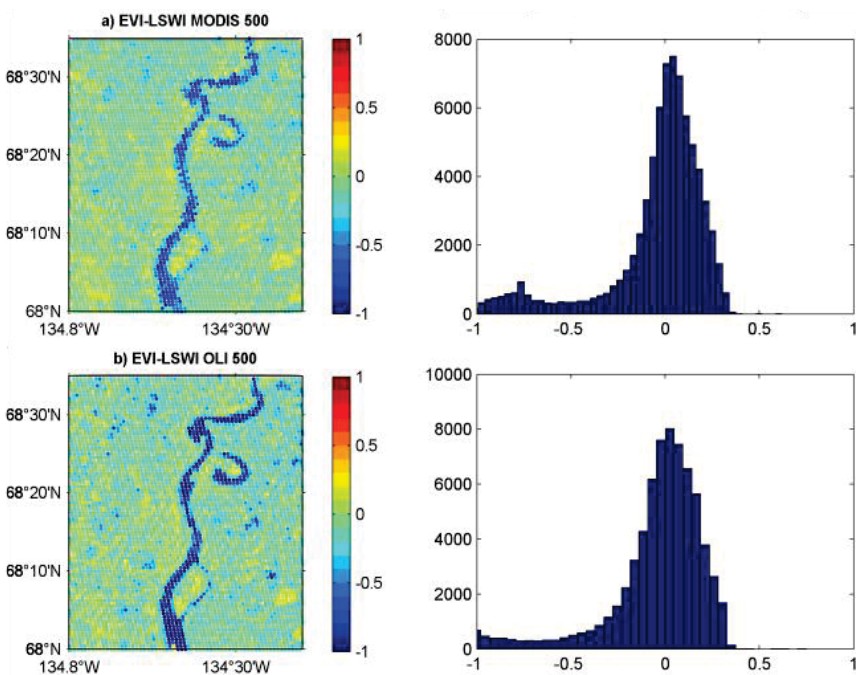

15    **Figure 4: Map of difference between EVI and LSWI indices and associated histogram in July 2013 for (a) MODIS and (b) OLI 500**





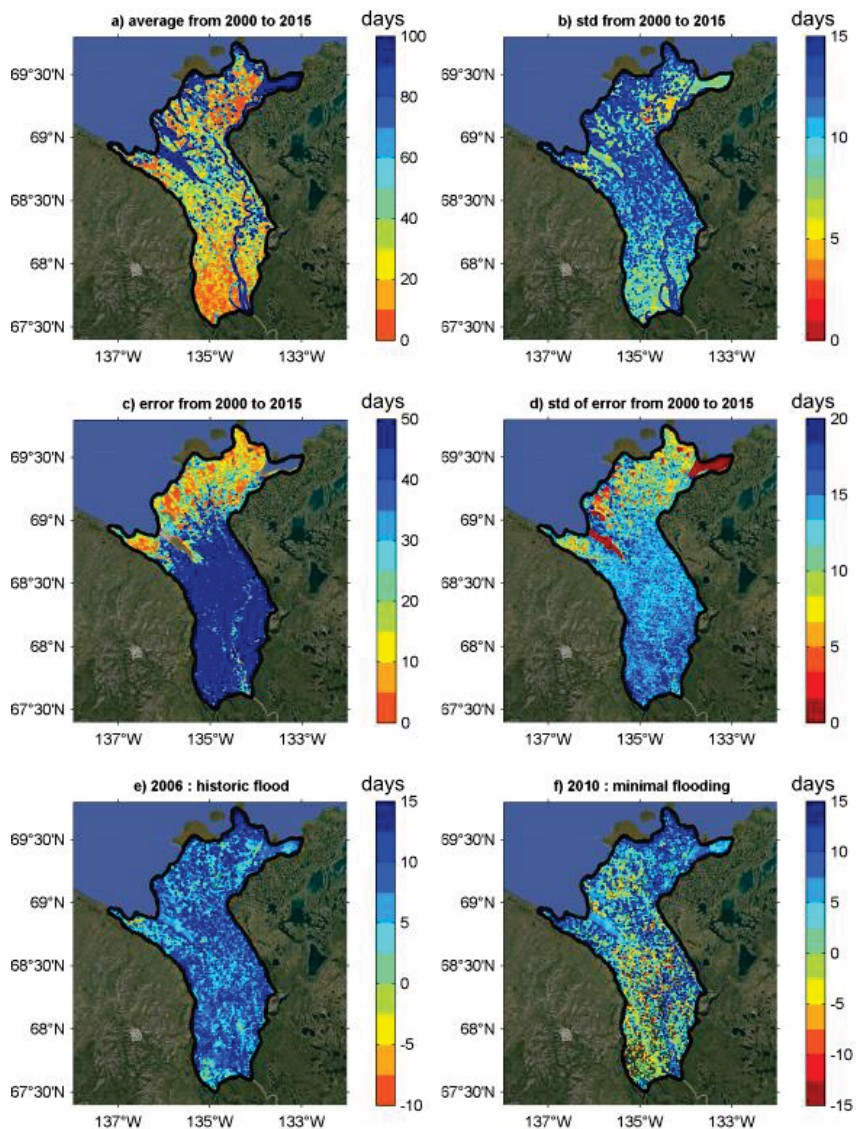

**Figure 5: Maps of surface water extent duration for (a) annual average from 2000 to 2015, (b) annual standard deviation from 2000 to 2015, (c) error average from 2000 to 2015, (d) standard deviation of error from 2000 to 2015, difference between annual average land water surface duration from 2000 to 2015 and land water surface duration during (e) 2006 associated with the highest flood event, and (f) 2010 associated with the lowest flood event recorded over the period.**



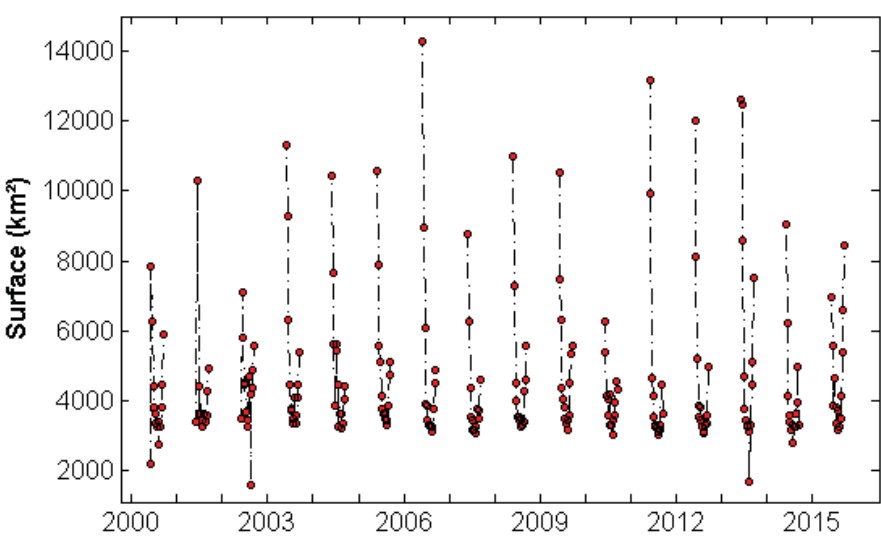

10    **Figure 6: Time series of surface water extent from 2000 to 2015, between June and September, derived from the MODIS images.**





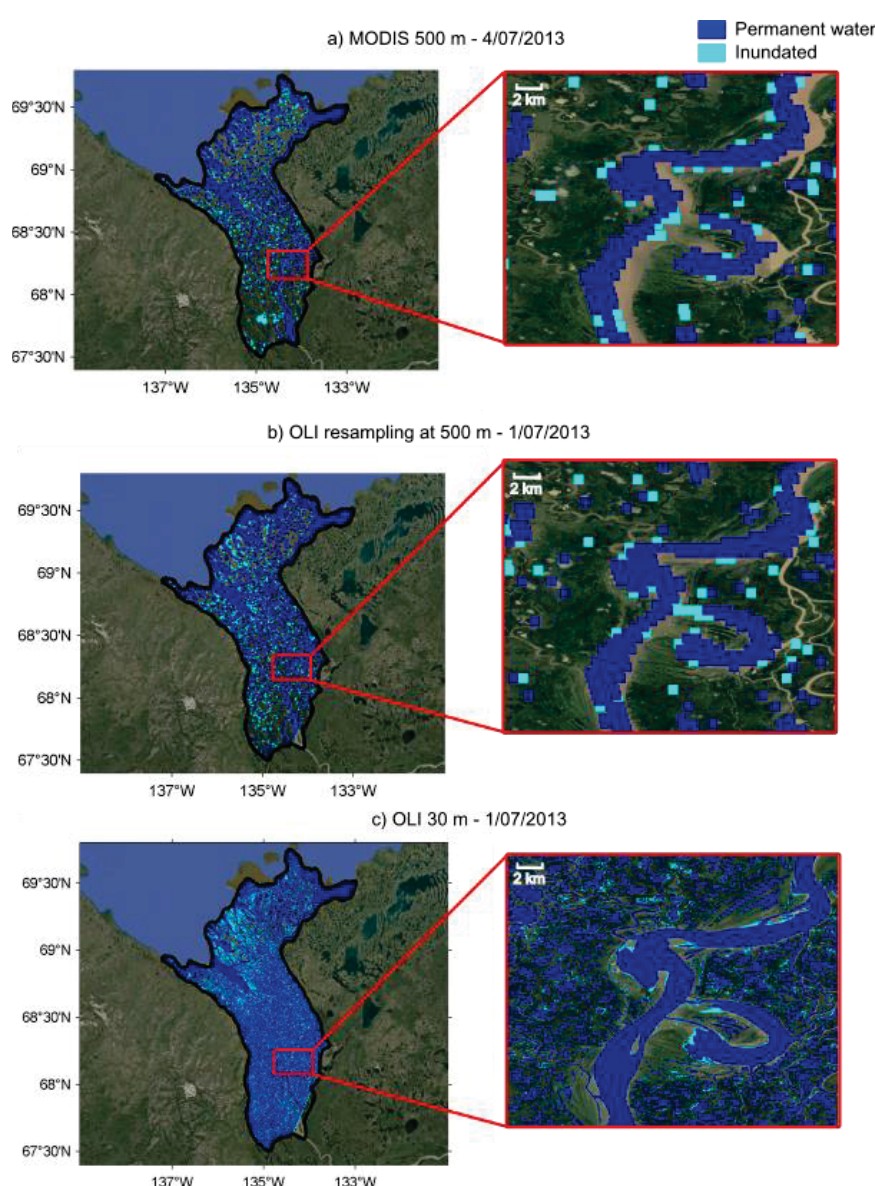

**Figure 7: Surface water extent estimated in July 2013 using (a) MODIS at 500 m (b) OLI at 500 m and (c) OLI at 30 m.**
5    **Permanent water bodies appear in blue and inundated areas in light blue.**





5  **Table 1: Validation of surface water extents (km²) determined using OLI 30 m, OLI 500 m, and MODIS 500 m images with Emmerton et al. (2007)**

|  | MODIS : 04/07/2013 OLI : 01/07/2013 | MODIS : 05/08/2013 OLI : 02/08/2013 |
| --- | --- | --- |
| MODIS 500 m | 3,798 | 3,298 |
| OLI 500 m | 4,499 | 3,859 |
| Emmerton et al., 2007 (channels+wetlands, km²) | 3,358 | 3,358 |
| Difference between MODIS 500 and Emmerton et al., 2007 | 440 km² (13 %) | 60 km² (2 %) |
| Difference between OLI 500 and Emmerton et al., 2007 | 1,141 (34 %) | 500 (15 %) |
| OLI 30 m | 7,685 | 7,156 |
| Emmerton et al., 2007 (channels+lakes+wetlands, km²) | 6,689 | 6,689 |
| Difference between OLI 30 and Emmerton et al., 2007 | 996 km² (13 %) | 467 km² (7 %) |





5    **Table 2: Satellite validation of surface water extent using OLI 30, OLI 500 and MODIS 500 m.**

| Date | MODIS : 04/07/2013 OLI : 01/07/2013 | MODIS : 05/08/2013 OLI : 02/08/2013 |
|---|---|---|
| **Permanent water MODIS (km²)** | 3,167 | 2,885 |
| **Permanent water OLI 500 (km²)** | 3,809 | 3,809 |
| **Permanent water OLI 30 (km²)** | 7,058 | 7,058 |
| **Inundated surfaces MODIS (km²)** | 577 | 250 |
| **Inundated surfaces OLI 500 (km²)** | 690 | 50 |
| **Inundated surfaces OLI 30 (km²)** | 627 | 98 |





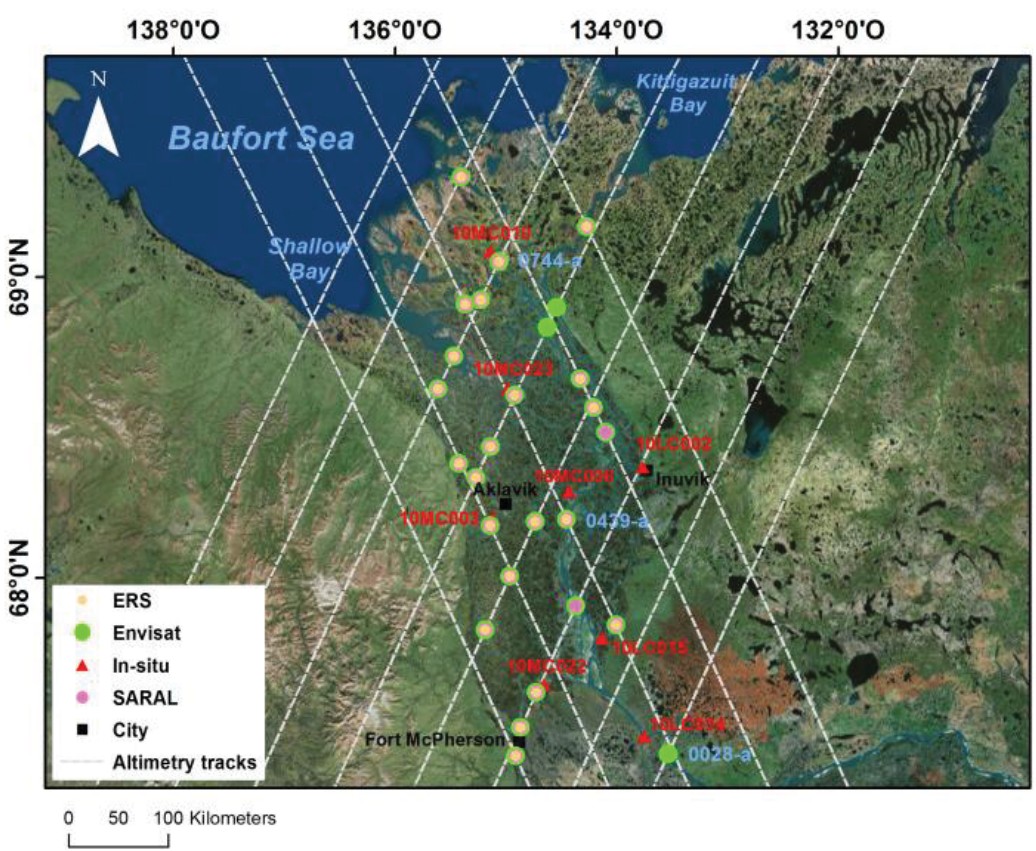

5   **Figure 8: Locations of virtual stations (VS) in the Mackenzie Delta for ERS-2 (yellow dots), Envisat (green dots) and SARAL (purple dots) altimetry missions. Altimetry tracks appear in grey. *In situ* stations are represented using red triangles.**





**Table 3: Statistic parameters obtained between altimetry-based water levels from ERS-2 and *in situ* water levels**

| Virtual station (SV) | In situ station | Distance (km) | River width at VS (m) | N | r | RMS (m) | R² | Bias (m) | Bias ICESat (m) |
|---|---|---|---|---|---|---|---|---|---|
| ERS-2-0439-a | 10MC008 | 11.44 | 1950 | 5 | 0.76 | 0.5 | 0.58 | 0.55 | 1.36 |
| ERS-2-0983-c | 10MC003 | 3.1 | 360 | 20 | 0.69 | 0.7 | 0.47 | - | - |
| ERS-2-0114-c | 10MC022 | 1.9 | 430 | 14 | -0.38 | 2.82 | 0.14 | - | - |
| ERS-2-0200-d | 10MC023 | 4.11 | 630 | 17 | 0.08 | 4.3 | 0 | - | - |
| ERS-2-0744-a | 10MC010 | 7.2 | 380 | 5 | 0.88 | 0.1 | 0.77 | - | -1.28 |
| ERS-2-0439-d | 10LC015 | 7.2 | 380 | 20 | 0.92 | 0.83 | 0.86 | - | - |



**Table 4: Statistic parameters obtained between altimetry-based water levels from ENVISAT and *in situ* water levels**

| Virtual station (SV) | In situ station | Distance (km) | River width at VS (m) | N | r | RMS (m) | R² | Bias (m) | Bias ICESat (m) |
|---|---|---|---|---|---|---|---|---|---|
| ENV-0439-a | 10MC008 | 11.44 | 1950 | 24 | 0.89 | 0.5 | 0.81 | 0.15 | 0.65 |
| ENV-0983-c | 10MC003 | 3.1 | 360 | 26 | 0.66 | 0.89 | 0.44 | - | - |
| ENV-0114-c | 10MC022 | 1.9 | 430 | 23 | 0.8 | 1.17 | 0.64 | - | - |
| ENV-0200-d | 10MC023 | 4.11 | 630 | 22 | 0.87 | 0.33 | 0.75 | - | - |
| ENV-0525-a | 10MC002 | 16.31 | 500 | 29 | 0.77 | 1.45 | 0.6 | - | - |
| ENV-0744-a | 10MC010 | 5.16 | 850 | 24 | 0.93 | 0.15 | 0.87 | - | -1.17 |
| ENV-0028-a | 10LC014 | 16.05 | 1360 | 17 | 0.83 | 1.84 | 0.7 | - | 2.35 |
| ENV-0439-d | 10LC015 | 7.2 | 380 | 28 | 0.65 | 1.75 | 0.43 | - | - |





**Table 5: Statistics parameters obtained between altimetry-based water levels from SARAL and *in situ* water levels**

| Virtual station (SV) | In situ station | Distance (km) | River width at VS (m) | N | r | RMS (m) | R² | Bias (m) | Bias ICESat (m) |
|---|---|---|---|---|---|---|---|---|---|
| SARAL-0439-a | 10MC008 | 11.44 | 1950 | 8 | 0.96 | 0.35 | 0.93 | -0.95 | -0.15 |
| SARAL-0983-c | 10MC003 | 3.1 | 360 | 6 | 0.9 | 0.4 | 0.8 | - | - |
| SARAL-0114-c | 10MC022 | 1.9 | 430 | 7 | 0.14 | 0.73 | 0.02 | - | - |
| SARAL-0200-d | 10MC023 | 4.11 | 630 | 6 | 0.76 | 0.3 | 0.57 | - | - |
| SARAL-0744-a | 10MC010 | 5.16 | 850 | 2 | 0.99 | 0.15 | 0.99 | - | -2.19 |
| SARAL-0439-d | 10LC015 | 7.2 | 380 | 5 | 0.95 | 1.3 | 0.9 | - | - |





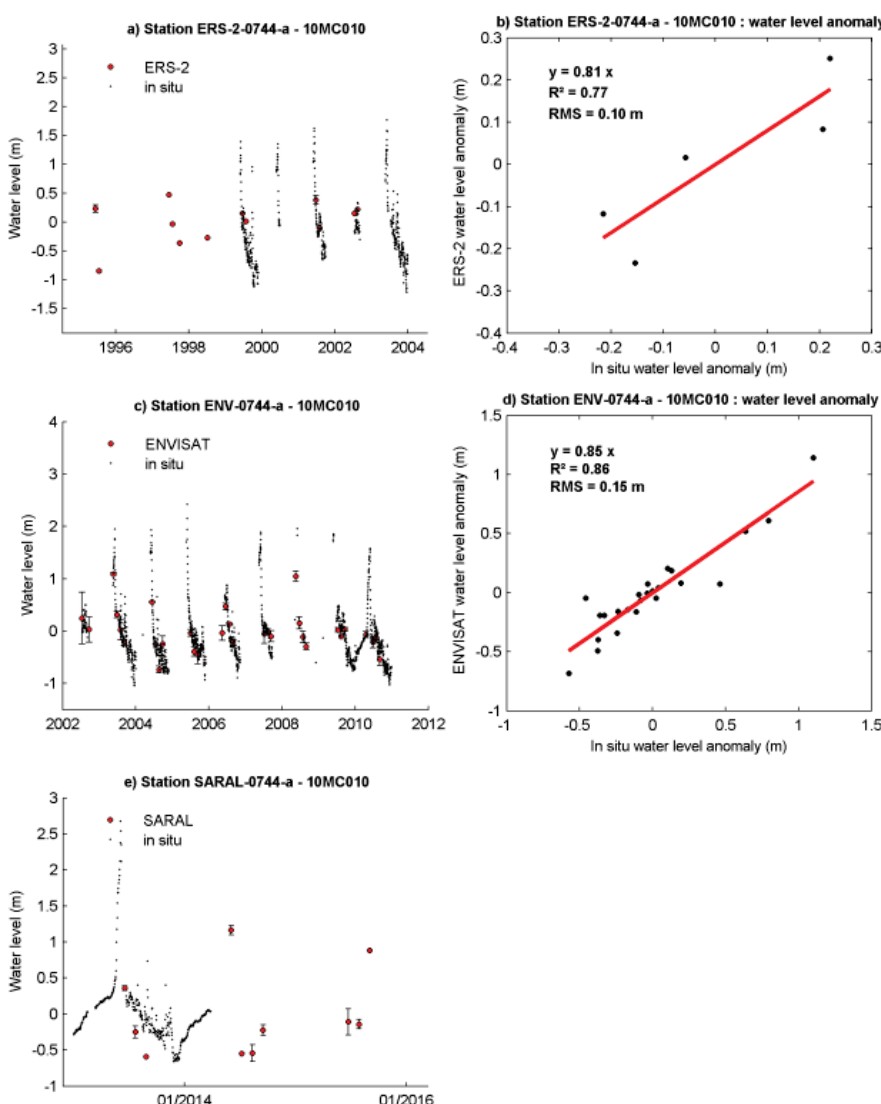

**Figure 9: Altimetry-based water levels from 1995 to 2015 compared with in situ water levels for the station 0744-a located in the downstream part in the Mackenzie Delta (a) using ERS-2 mission and (b) water level anomaly with statistic parameters, (c) using ENVISAT mission and (d) water level anomaly with statistic parameters and (e) using SARAL mission**



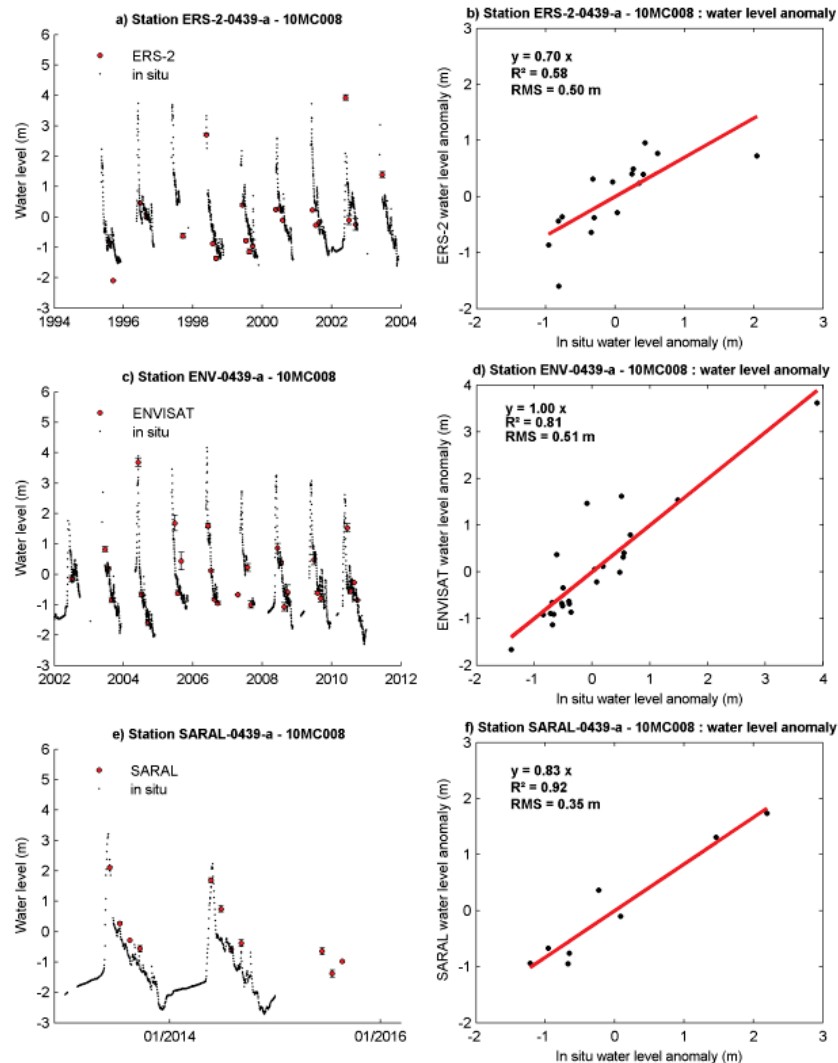

**Figure 10: Altimetry-based water levels from 1995 to 2015 compared with in situ water levels for the station 0439-a located in the center in the Mackenzie Delta (a) using ERS-2 mission and (b) water level anomaly with statistic parameters, (c) using ENVISAT mission and (d) water level anomaly with statistic parameters, (e) using SARAL mission and (f) water level anomaly with statistic parameters**





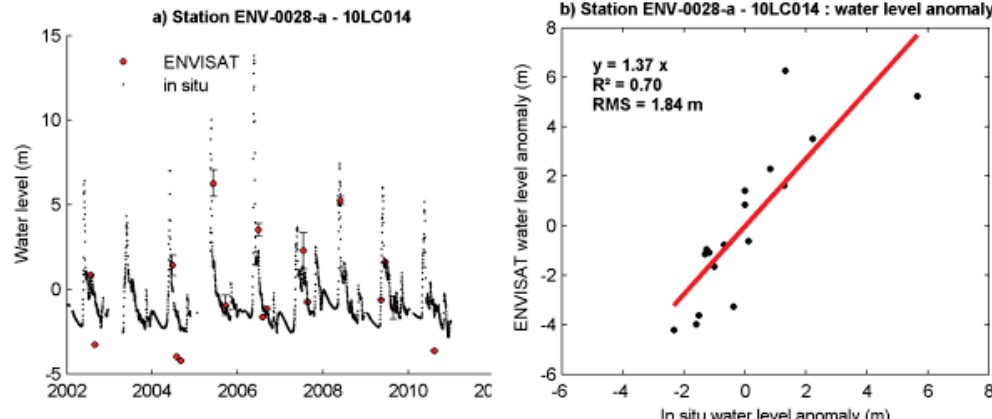

**Figure 11: Altimetry-based water levels from 2002 to 2010 compared to in situ water levels for the station 0439-a located in the center in the Mackenzie Delta (a) using ENVISAT mission and (b) water level anomaly with statistic parameters**



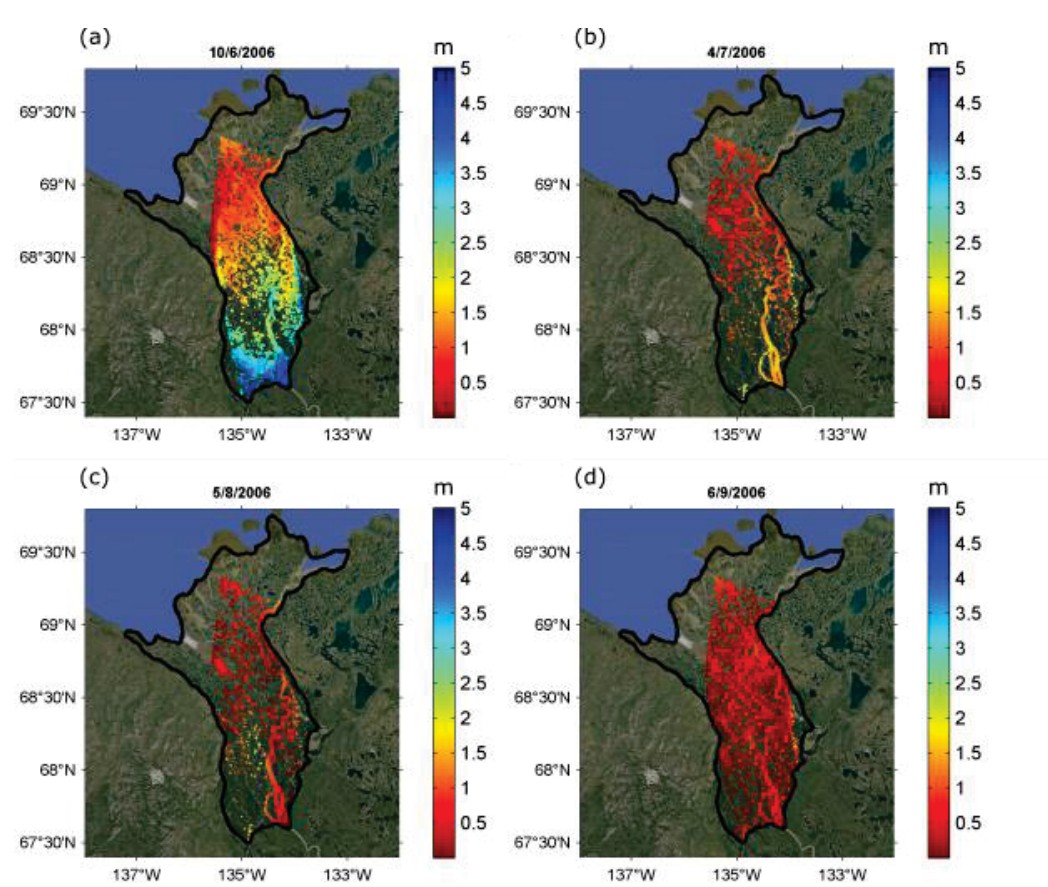

**Figure 12: Water level maps in the Mackenzie Delta in 2006 (historic flood) obtained combining inundated surfaces determined using MODIS images with altimetry-derived water levels (a) in June, (b) in July, (c) in August and (d) in September**





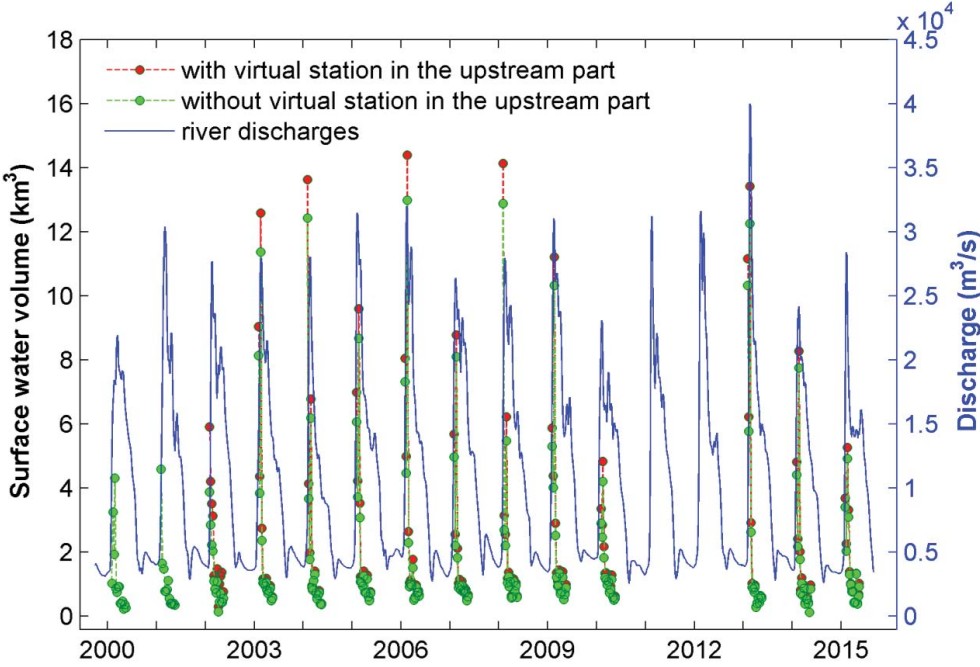

**Figure 13: Surface water volume from 2000 to 2015, determined by combining inundated surfaces from MODIS with altimetry data. 167 red points correspond to surface water volume obtained with a virtual station located in the upstream part of the Delta, green points to surface water volume without a virtual station located in the upstream part of the Delta. The Mackenzie River Delta**
10  **discharges at 10LC014 gauge station appear in blue.**