# Peer review of "Quantification of surface water volume changes in the Mackenzie Delta using satellite multi-mission data"

_Hydrology and Earth System Sciences, 2017_

## Referee Comment (RC1) · Anonymous Referee #1 · 2 Nov 2017

This manuscript quantifies temporal changes of surface water volume water storage in the Mackenzie Delta based on multispectral images and altimetry data. The authors validates (1) classification of land water surface with multispectral images, (2) water level estimates by altimetry data, and (2) surface volume estimations retrieved by both multispectral and altimetry data. The manuscript is well-written and easy to follow their methodology and results. However, I am compelled to say that the present manuscript misses to demonstrate the scientific significance to stand alone in a HESS's publication.

The authors fail to demonstrate its originality of the manuscript. The authors described

"the originality and novelty ... (P3L7)". However, I felt that the present manuscript just applied existing approaches proposed by Frappart et al. (2006b, 2010, 2012) for long-time period in the target area. I could not understand challenges and difficulties in the present manuscript. I understand that the authors processed a number of data carefully and correctly. However, scientific paper needs to demonstrate (1) scientific questions or challenges that present human being does not know/understand, (2) to propose how to solve the issue (i.e., hypothesis) and (3) discuss to differentiate its originality from existing studies. I suggest the authors to reconstruct the manuscript again to demonstrate its originality. The present manuscript is quite good as engineering/technical description paper, but needs originality as a scientific paper.

[Other Issues] 1. P2L32: What are traditional methods? 2. Section 4.1 (P7L1-L6): How did authors decide the criteria? 3. P9L1: Please describe the definition of the "errors". 4. P9L35: It is better to explain the method of Emmerton et al. (2007) since the authors used Emmerton et al.'s results for the validation. 5. I recommend the authors to discuss generality of their approach. Namely, what kinds of difficulty do you expect if the other researchers would apply the same method for other areas?

---

## Referee Comment (RC2) · Anonymous Referee #2 · 4 Nov 2017

Normandin et al. used multi satellite sensor integration (multispectral and radar altimetry) to quantify both surface water extent and volume dynamics across the Mackenzie Delta for a time period of 15 years. The information (time series) generated in this study is of high relevance for many applications and the methodology used is appropriate and well documented/ described. Although this is not the first study to combine satellite altimetry with remotely sensed surface water extents for water volume estimations, the application of these methods over a large and complex river delta and a 15-year time period makes this study a significant contribution to the field and of great interest to the HESS community. Nevertheless, the manuscript is rather premature and not suitable for publication in its present form. The authors fail to clearly distinguish

their work from work that has already been done and the significance and novelty of their research isn't presented in its full potential. In addition, the authors leave it up to the readers to find the key results and highlights of the research. Rather than conveying the findings in a limited number of carefully designed figures and tables, the authors present an abundance of material that makes it difficult for the reader to ingest and enjoy the paper. The writing of the paper is often in the style of a technical report and the manuscript lacks flow in the argumentation and a proper discussion section, where the limitations and implications of the research are discussed in detail. Due to these issues, the manuscript is not suitable for publication in HESS in its present form and needs to undergo major revisions before it can be considered suitable for publication.

Specific Comments: The number of Figures and Tables is very high in relation to the information and novelty content of the manuscript. I highly recommend to reduce the number of Figures so that only the key information and results is presented. For example, the statistics presented in three separate tables 3,4 and 5 should be easy enough to show in a single graph, which would also make it much easier for the reader to get the main points without having to search. The same applies to the abundance of inundation extent and duration maps that are shown. Introduction: line 7 to 23: This sounds like the study area section (which should go to methods). page 2 line 29: Thus, the understanding of these dynamic environments is a societal and scientific stake to anticipate and manage 30 their evolutions at medium and long term time scales. This is confusing and I do not follow what your argument is here. Consider re-writing and clearly making your point. page 2 line 33: I agree with you that they probably are the only way but I would be carefull with this statement, considering that large-scale 3-d hydrodynamic modeling is getting more and more powerful and feasible. Also airborne remote sensing is an alternative. Consider rewriting. page 3 line 4: use quantify surface water extents instead of "spatial extent of surface water extents" page 3 line 8: The whole intro is poorly structured and lacks argumentative flow, which makes it rather difficult to read. It reads a bit like a "staccato" listing of important but often slightly unrelated and repetitive pieces of information. Consider rewriting the intro with
improved flow and less repetitions and focus on the a) background, b) significance and innovation and c) motivation of your research, rather than a very detailed description of your study area and corresponding environmental processes. You should clearly state that the sensors that you integrated haven't been integrated in this way for this particular quantification (if this is the case) and add more emphasis on the usefulness of the information about surface water extent and volume that you are generating in this study (which is certainly of very high importance). What can a time series of surface water volume be used for (i.e. studying climate feedback or sea level rise...). page 3 line 6: Where you cite papers that have "successfully applied this approach": You should describe the very closely related studies with more detail and then highlight what your study adds to this existing body of knowledge. line 36. Given the very high relevance of existing large scale surface water extent time series for your study (you could have used some of those for validation), you may want to consider to cite the state-of-the-art literature here: Pekel, J., A. Cottam, N. Gorelick, and A. S. Belward (2016), High-resolution mapping of global surface water and its long-term changes, Nature, 540, 418–422, doi:10.1038/nature20584. Tulbure, M. G., M. Broich, S. V Stehman, and A. Kommareddy (2016), Surface water extent dynamics from three decades of seasonally continuous Landsat time series at subcontinental scale in a semi-arid region, Remote Sens. Environ., 178, 142–157, doi:10.1016/j.rse.2016.02.034. Klein, I., Gessner, U., Dietz, A.J., Kuenzer, C., 2017. Global WaterPack – A 250 m resolution dataset revealing the daily dynamics of global inland water bodies. Remote Sens. Environ. 198, 345–362. doi:10.1016/j.rse.2017.06.045 page 3 line 10: In my opinion, the study region section is unnecessarily lengthy. Consider focusing only on the information that is relevant to your study and methods. Consider improving the reading flow by connecting sentences with similar information. page 4 line 6: Are you sure it's "raw radiance"? You mention further down it's surface reflectance. page 4 line 25: It is true that there will be plenty of data gaps in the OLI time series but what is the time step that you require for the surface water extent dynamics in your study? How quickly does the extent change over time? What about Sentinel-2? page 6 line 4: You may want to consider Tulbure,

Klein and Pekel 2016 and mention that the state of the art for this type of classification is machine learning but that you chose a simplified spectral indices approach because... page 6 line 31: You think one OLI image is enough for cross-validation? page 7 line 8: you mean $\frac{2}{3}$ of the "annual" study period from June to September? Your study period is 2000-2015 right? page 8 line 6: What is rough delineation of cross sections? Do you have DEM data or do you just refer to location of rivers? page 8 line 9: What is the refined process? page12 line 24: The correlation between discharge and surface water volumes should be discussed more here. Due to the size of the delta, there would be significant lag effects (time that the flow takes to pass through the delta) that are not captured by a simple correlation. You may want to consider to have a look at similar type of models and discuss your work in that context (i.e. Heimhuber, V., Tulbure, M.G., Broich, M., 2017. Modeling multidecadal surface water inundation dynamics and key drivers on large river basin scale using multiple time series of Earth-observation and river flow data. Water Resour. Res. 53, 1–19. doi:10.1002/2016WR019858). The relationship between discharge and water volume is unlikely to be linear so an r2 of 0.66 is pretty high for your application - I recommend discussing. Figure 3: A scale bar would make it easier to get an idea of the size of the index images that are shown in each panel. Figure 5: You might want to keep the direction of your colour bar consistent (i.e. more days is red, less days is blue). page 13 line 6 & Figure 6: Given that the surface water extent of the first time step of your annual time period is always the highest, maybe you should have started the annual time period 1 month earlier, to ensure that you always capture the peak of the flood extent. I find it problematic that you state that surface water extent is maximum in June, given that you never looked at May. Figure 7: How about overlaying the classified 500m pixel MODIS and Landsat image to highlight the differences. Give pixels where both agree one colour and then another colour for only water on Landsat and one for only water on MODIS.

Technical Corrections: page 1, line 20: In this study, the dynamics of surface water extent and volume "are or were" analyzed from 2000 to 2015 by combining multi-satellite information from MODIS multispectral images at 500 m spatial resolution and river

stages derived from ERS-2 (1995-2003), ENVISAT (2002-2010) and SARAL (since 2013) altimetry data. page 2 Line 5: Discharge instead of discharger page 7 line 10: you mean Hereafter? page 13 line 11: "." missing after MODIS? page 13 line 20: mistake in "Frthat" page 13 line 23: Sentence starting with "Besides, these..." is gramatically wrong and I do not get the point.

---

## Author Comment (AC1) · 1 Dec 2017

R.C.: Reviewer's Comment A.R.: Author's Reponse

Referee #1 This manuscript quantifites temporal changes of surface water volume water storage in the Mackenzie Delta based on multispectral images and altimetry data. The authors validates (1) classification of land water surface with multispectral images, (2) water level estimates by altimetry data, and (2) surface volume estimations retrieved by both multispectral and altimetry data. The manuscript is well-written and easy to follow their methodology and results. However, I am compelled to say that the present manuscript misses to demonstrate the scientific significance to stand

alone in a HESS's publication. The authors fail to demonstrate its originality of the manuscript. The authors described "the originality and novelty ... (P3L7)".However, I felt that the present manuscript just applied existing approaches proposed by Frappart et al. (2006b, 2010, 2012) for long-time period in the target area. I could not understand challenges and difficulties in the present manuscript. I understand that the authors processed a number of data carefully and correctly. However, scientific paper needs to demonstrate (1) scientific questions or challenges that present human being does not know/understand, (2) to propose how to solve the issue (i.e., hypothesis) and (3) discuss to differentiate its originality from existing studies. I suggest the authors to reconstruct the manuscript again to demonstrate its originality. The present manuscript is quite good as engineering/technical description paper, but needs originality as a scientific paper.

We thank Referee 1 to offer us the opportunity to improve our manuscript (supplement information joined here). In the corrected version of the manuscript, we detailed in what our methodology differs from what we published before in other large-scale basins: "In the past, this approach has been applied in tropical (e.g., the Amazon (Frappart et al., 2012), Mekong (Frappart et al., 2006b)) and peri-Arctic (e.g. the Lower Ob' basin, (Frappart et al., 2010) major river basins allowing to provide direct observations of the spatio-temporal dynamics of surface water storage. Several limitations prevent them to be used over estuaries and deltas. The first is the too coarse spatial resolution of the datasets used for retrieving the flood extent that ranges from 1 km with SPOT-VGT images used in the Lower Mekong Basin to $\sim 0.25°$ with the Global Inundation Extent from Multi-Satellite (GIEMS, Papa et al., 2010) for the Lower Ob' and the Amazon basins. The second is inherent to the datasets used in these studies. For the Mekong Basin, due to the small number of available spectral bands present in the VGT sensor, a mere threshold on NDVI was applied. For the Amazon and the Lower Ob', as GIEMS dataset is using surface temperatures from SSM/I, no valid data are available at less than 50 km from the coast. The originality and novelty of the study is the use of multi-space mission data at medium spatial, temporal and spectral resolutions to monitor

surface water storage changes in a deltaic environment over a fifteen-year time period."

We also explained which scientific questions motivated our study: "Earlier studies pointed out i) the lack of continuous information in the Mackenzie delta to study the spatial distribution of water levels during the flood events and to analyze the relationship between flood severity and the timing and duration of break-up in the delta (Goulding et al, 2009b), ii) the importance of the tributaries to the Mackenzie River (i.e., Peel and Arctic Red rivers) on break-up and ice-jam flooding in the delta (Goulding et al., 2009a). As the goal of this study is to characterize the spatio-temporal dynamics of surface water, both in surface and storage, in the Mackenzie delta, north west territories of Canada, in response to spring ice break-up and snow melt, over the period 2000-2015, it will provide important new information for a better understanding of the hydro-climatology of the region." We widely modified the structure of the manuscript to put the stress on the scientific results. We added a supplementary information file for the technical aspects. We strengthened the introduction and conclusion on the interest of our study for the hydro-climatological community. We divided in the former version section 5 (results and discussion) in 2 separated sections: the results (section 5) and the discussion (section 6). You will find our detailed answers to your comments below.

[OtherIssues] RC 1: page 2 line 32: What are traditional methods? A.R 1: We meant networks of in-situ gauge stations that are insufficiently dense in this region for the monitoring of the wetlands hydrodynamics although denser than in many regions of the world thanks to the efforts of the Canada Water Office to provide a good monitoring of Canadian rivers and lakes. We replaced this sentence with: "However, it is nearly impossible to provide a long-term monitoring with traditional methods using in-situ measurements in such a large and heterogeneous environment. Satellite remote sensing methods are the only way to solve this problem offers a unique opportunity for the continuous observation of wetlands and floodplains".

RC 2: page 7 – line 1-7 (Section4.1): How did authors decide the criteria? A.R 2: We applied the approach proposed by Sakamoto et al., (2007). In this method, as

explained in our manuscript, pixels are considered as water-related pixels if: - CASE 1: EVI-LSWI<=0.05 and EVI<0.3 - CASE 2: EVI <= 0.05 and LSWI <= 0 In our study, for the case 2, we only use EVI <=0.05 since no negative values were found for images (Figure S1). This approach was validated through comparison against the few available Landsat 8 images over our study area.

RC 3: page 9 – line 1: Please describe the defiŗnition of the "errors". A.R 3: We better defined the errors in the manuscript. For the figure 3a, surface extent is calculated using Sakamoto et al., (2007) classification: 0 = vegetation, 1 = permanent water, 2 = inundated pixel and 3 = mixed pixel. Only classes 1 and 2 are used for the figure 5a. Errors are calculated using the mixed pixels, corresponding to the class 3. This explanation has been added in the manuscript as follow: "Following Sakamoto et al. (2007) method, all pixels of 8-day image have been classified into 4 classes: class 0 corresponding to vegetation, class 1 to permanent water, class 2 to inundation and class 3 to mixture of land and water. Map of annual average of land water surface, composed of inundated and permanent water bodies, was obtained at spatial and temporal resolutions of 500 m and 8 days respectively from June to September over the 2000-2015 period (Figure 3a) using classes 1 and 2." ... "Maps of errors made on land water surface duration with associated standard deviation are shown in Figure 3c and 3d over 2000-2015. Errors on land water surface duration are calculated using mixed pixels, corresponding to the class 3 in Sakamoto et al., (2007) classification. Standard deviation of error is presented in Figure 3d."

RC 4: page 9 line 35: It is better to explain the method of Emmerton e tal. (2007) since the authors used Emmerton et al.'s results for the validation. A.R 4: The following sentence has been added to the manuscript: Emmerton et al., (2007) classified the Mackenzie Delta habitat in lakes, channels, wetlands and dry floodplains using information from a topographic maps derived from aerial photographies taken during the 1950's for low water periods.

RC 5: I recommend the authors to discuss generality of their approach. Namely,

what kinds of difiñǍculty do you expect if the other researchers would apply the same method for other areas? A.R 5: In our opinion, this approach can be applied in any other deltaic and estuarine environments as MODIS and altimetry data are available globally. We added the following sentence in the conclusion: "This approach can be applied to any other deltaic and estuarine environments as MODIS and altimetry data are available globally. The major limitations are i) the presence of clouds and dense vegetation cover that prevent the use of MODIS images, ii) the relatively coarse spatial resolution of MODIS images, iii) the coarse coverage of altimetry tracks. They can be overcome i) using SAR images for flood extent monitoring as Frappart et al. (2005), ii) using images with a higher spatial resolution, iii) combing information the different altimetry missions orbiting simultaneously. The recent launches of Sentinel-1, Sentinel-2 and 3 offer new opportunities for flood extent monitoring at higher spatial (from ∼10 m to 300 m) and temporal (a few days) resolutions".

Please also note the supplement to this comment:
https://www.hydrol-earth-syst-sci-discuss.net/hess-2017-170/hess-2017-170-AC1-supplement.pdf
* * *

---

## Author Comment (AC2) · 1 Dec 2017

R.C.: Reviewer's Comment A.R.: Author's Reponse

Referee #2 Normandin et al. used multi satellite sensor integration (multispectral and radar altimetry) to quantify both surface water extent and volume dynamics across the Mackenzie Delta for a time period of 15 years. The information (time series) generated in this study is of high relevance for many applications and the methodology used is appropriate and well documented/ described. Although this is not the first study to combine satellite altimetry with remotely sensed surface water extents for water volume estimations, the application of these methods over a large and complex river delta

and a 15-year time period makes this study a significant contribution to the field and of great interest to the HESS community. Nevertheless, the manuscript is rather premature and not suitable for publication in its present form. The authors fail to clearly distinguish their work from work that has already been done and the significance and novelty of their research isn't presented in its full potential. In addition, the authors leave it up to the readers to find the key results and highlights of the research. Rather than conveying the findings in a limited number of carefully designed figures and tables, the authors present an abundance of material that makes it difificult for the reader to ingest and enjoy the paper. The writing of the paper is often in the style of a technical report and the manuscript lacks flow in the argumentation and a proper discussion section, where the limitations and implications of the research are discussed in detail. Due to these issues, the manuscript is not suitable for publication in HESS in its present form and needs to undergo major revisions before it can be considered suitable for publication.

We thank the Referee for his helpful comments that helped us in improving our manuscript. We widely modified the structure of the manuscript to put the stress on the scientific results. We added a supplementary information file for the technical aspects. We strengthened the introduction and conclusion on the interest of our study for the hydro-climatological community. We divided in the former version section 5 (results and discussion) in 2 separated sections: the results (section 5) and the discussion (section 6). You will find our detailed answers to your comments below. You will find our detailed answers to your comments below.

Specific Comments: RC 1: The number of Figures and Tables is very high in relation to the information and novelty content of the manuscript. I highly recommend to reduce the number of Figures so that only the key information and results is presented. For example, the statistics presented in three separate tables 3,4 and 5 should be easy enough to show in a single graph, which would also make it much easier for the reader to get the main points without having to search. The same applies to the abundance

of inundation extent and duration maps that are shown. A.R 1: Following your suggestion, we reduced the number of Tables and Figures. The statistics of the validation of altimetry-based water levels were merged from three to one table. A similar merging was also applied for inundation extent. Former figures 3, 4 and 7 were moved to the supplementary information document.

RC 2: Introduction: line 7 to 23: This sounds like the study area section (which should go to methods). A.R 2: This paragraph was merged with the existing content of the study area section (that was reduced).

RC 3: page 2 line 29: Thus, the understanding of these dynamic environments is a societal and scientific stake to anticipate and manage their evolutions at medium and long term time scales. This is confusing and I do not follow what your argument is here. Consider re-writing and clearly making your point. A.R 3: We rewrote as follows: "Improving our knowledge on the dynamics of the surface water reservoir in circumpolar areas is crucial for a better understanding of their role in flood hazard, carbon production, greenhouse gases emission, sediment transport, exchange of nutrients and land-atmosphere interactions".

RC 4: page 2 line 33: I agree with you that they probably are the only way but I would be carefull with this statement, considering that large-scale 3-d hydrodynamic modeling is getting more and more powerful and feasible. Also airborne remote sensing is an alternative. Consider rewriting. A.R 4: To focus on the long-term monitoring, we rewrote as follows: "Mapping surface water extent at the Mackenzie Delta scale is an important issue. However, it is nearly impossible to provide a long-term monitoring using in-situ measurements in such a large and heterogeneous environment. Satellite remote sensing methods offers a unique opportunity for the continuous observation of wetlands and floodplains".

RC 5: page 3 line 4: use quantify surface water extents instead of "spatial extent of surface water extents" A.R 5: Corrected.

RC 6: page 3 line 8: The whole intro is poorly structured and lacks argumentative flow, which makes it rather difficult to read. It reads a bit like a "staccato" listing of important but often slightly unrelated and repetitive pieces of information. Consider rewriting the intro with improved flow and less repetitions and focus on the a) background, b) signïficance and innovation and c) motivation of your research, rather than a very detailed description of your study area and corresponding environmental processes. You should clearly state that the sensors that you integrated haven't been integrated in this way for this particular quantification (if this is the case) and add more emphasis on the usefulness of the information about surface water extent and volume that you are generating in this study (which is certainly of very high importance). What can a time series of surface water volume be used for (i.e. studying climate feedback or sea level rise...). A.R 6: We completely restructured the introduction as you mentioned. We strengthened the significance and innovation (see our response to your comment below) and the motivation for this study as follows: "Earlier studies pointed out i) the lack of continuous information in the Mackenzie delta to study the spatial distribution of water levels during the flood events and to analyze the relationship between flood severity and the timing and duration of break-up in the delta (Goulding et al, 2009b), ii) the importance of the tributaries to the Mackenzie River (i.e., Peel and Arctic Red rivers) on break-up and ice-jam flooding in the delta (Goulding et al., 2009a). As the goal of this study is to characterize the spatio-temporal dynamics of surface water, both in surface and storage, in the Mackenzie delta, north west territories of Canada, in response to spring ice break-up and snow melt, over the period 2000-2015, it will provide important new information for a better understanding of the hydro-climatology of the region".

RC 7: page 3 line 6: Where you cite papers that have "successfully applied this approach": You should describe the very closely related studies with more detail and then highlight what your study adds to this existing body of knowledge. A.R 7: We added the following sentences: " In the past, this approach has been applied in tropical (e.g., the Amazon (Frappart et al., 2012), Mekong (Frappart et al., 2006b)) and peri-Arctic

(e.g. the Lower Ob' basin, (Frappart et al., 2010) major river basins allowing to provide direct observations of the spatio-temporal dynamics of surface water storage. Several limitations prevent them to be used over estuaries and deltas. The first is the too coarse spatial resolution of the datasets used for retrieving the flood extent that ranges from 1 km with SPOT-VGT images used in the Lower Mekong Basin to $\sim 0.25°$ with the Global Inundation Extent from Multi-Satellite (GIEMS, Papa et al., 2010) for the Lower Ob' and the Amazon basins. The second is inherent to the datasets used in these studies. For the Mekong Basin, due to the small number of available spectral bands present in the VGT sensor, a mere threshold on NDVI was applied. For the Amazon and the Lower Ob', as GIEMS dataset is using surface temperatures from SSM/I, no valid data are available at less than 50 km from the coast. The originality and novelty of the study is the use of multi-space mission data at medium spatial and temporal resolutions to monitor surface water storage changes in a deltaic environment over a fifteen-year time period".

RC 8: page 3 line 36. Given the very high relevance of existing large scale surface water extent time series for your study (you could have used some of those for validation), you may want to consider to cite the state-of-the-art literature here: Pekel, J., A. Cottam, N. Gorelick, and A. S. Belward (2016), High resolution mapping of global surface water and its long-term changes, Nature, 540, 418–422, doi:10.1038/nature20584. Tulbure, M. G., M. Broich, S. V Stehman, and A. Kommareddy (2016), Surface water extent dynamics from three decades of seasonally continuous Landsat time series at subcontinental scale in a semi-arid region, Remote Sens. Environ., 178, 142–157, doi:10.1016/j.rse.2016.02.034. Klein, I., Gessner, U., Dietz, A.J., Kuenzer, C., 2017. Global WaterPack – A 250 m resolution dataset revealing the daily dynamics of global inland water bodies. Remote Sens. Environ. 198, 345–362. doi:10.1016/j.rse.2017.06.045 A.R 8: We agree on this comment. In fact, our manuscript was ready to submit in fall 2016 when we added the comparison with Landsat-8. We added these recent references.

[Figure]

RC 9: page 3 line 10: In my opinion, the study region section is unnecessarily lengthy. Consider focusing only on the information that is relevant to your study and methods. Conside improving the reading flow by connecting sentences with similar information. A.R 9: The section has been rewritten and is shorter taking, taking into account only important and relevant information.

RC 10: page 4 line 6: Are you sure it's "raw radiance"? You mention further down it's surface reflectance. A.R 10: Corrected, it was a mistake.

RC 11: page 4 line 25: It is true that there will be plenty of data gaps in the OLI time series but what is the time step that you require for the surface water extent dynamics in your study? How quickly does the extent change over time? What about Sentinel-2? A.R 11: Surface water storage dynamic happens really fast in this environment. This is why we chose to use images available every 8-day to be able to monitor the variations of surface water extent and storage. Until the launch of Landsat-8 in 2013, only 2 images were available for our study period (from June to September). Our study presents multi-year variations of surface water storage and extent. There are still too few Sentinel-2 images to allow a long-term monitoring. Besides, to our knowledge, surface reflectances from Sentinel-2 are not available in the Mackenzie, only top of atmosphere reflectances. We agree with this comment and we mentioned the interest of the Sentinel missions in the conclusion: "The recent launches of Sentinel-1, Sentinel-2 and 3 offer new opportunities for flood monitoring at higher spatial ($\sim$10 m) and temporal (a few days) resolutions".

RC 12: page 6 line 4: You may want to consider Tulbure, Klein and Pekel 2016 and mention that the state of the art for this type of classification is machine learning but that you chose a simplified spectral indices approach because... A.R 12: We totally agree you. Machine learning techniques need to have ground validation. This was not our case. This why we choose this approach. We added your comment to our manuscript: "As we do not have any external information to perform a supervised classification as the current state of the art machine learning techniques, we used the

approach proposed by (Sakamoto et al., 2007) to monitor the land water surface extent in the Mackenzie Delta (Figure 2)".

RC 13: page 6 line 31: You think one OLI image is enough for cross-validation? A.R 13: We used the only information available that consists of two Landsat-8 images. Two OLI images were used to validate (01/07/2013 and 02/08/2013) and validation results are shown in Table 1.

RC 14: page 7 line 8: you mean of the "annual" study period from June to September? Your study period is 2000-2015 right? A.R 14: Yes exactly, I've added annual. Yes my study period is 2000-2015.

RC 15: page 8 line 6: What is rough delineation of cross sections? Do you have DEM data or do you just refer to location of rivers? A.R 15: As explained in the manuscript and in the references cited, the first is a rough delineation of the cross-sections (typically plus or minus 5 km from the river banks) based on satellite images. The MAPS software allows to superimpose altimeter tracks to a Google Earth background. Then, the shape of the altimeter along-track profiles permit to identify the river that is generally materialized as a shape of "V" or "U" with the lower elevations corresponding to the water surface (see Santos da Silva et al., 2010 and Baup et al., 2014 for more details). We added this last sentence to the manuscript.

RC 16: page 8 line 9: What is the reffned process? A.R 16: We modified as follows: "Valid altimetry data were selected through a refined process that consists in eliminating outliers and measurements over non-water surfaces based on visual inspection".
RC 17: page12 line 24: The correlation between discharge and surface water volumes should be discussed more here. Due to the size of the delta, there would be significant lag effects (time that the flow takes to pass through the delta) that are not captured by a simple correlation. You may want to consider to have a look at similar type of models and discuss your work in that context (i.e. Heimhuber, V., Tulbure, M.G., Broich, M., 2017. Modeling multidecadal surface water inundation dynamics and key

drivers on large river basin scale using multiple time series of Earth-observation and river flow data. Water Resour. Res. 53, 1–19. doi:10.1002/2016WR019858). The relationship between discharge and water volume is unlikely to be linear so an r2 of 0.66 is pretty high for your application - I recommend discussing.

A.R 17: Thank very much for your comment. As you mentioned, the situation is quite similar in the Mackenzie delta. There are some floodplains connected and some other non- connected to the river. As it is mentioned in the manuscript, after the flood peak in June, the validation performed between MODIS and Landsat showed that MODIS only detects water over river channels and connected floodplains whereas the small non-connected lakes were not. We totally agree on the non-linearity between volume and discharge. We also noted it on the first study we published using this technique (Frappart et al., 2005). We computed cross-correlations between storage and discharge. We did not found any time-lag. This lack of time-lag is likely due to the time-step of 8 days of our estimates. We added the following paragraph in section spatio-temporal dynamics of surface water storage: "The comparison between storage and flux (discharge) exhibits a quite good correlation (R=0.66 with no time-lag) between these two quantities. Several studies demonstrated that there is no linear relationship between surface water extent, surface water volume and river discharge due to the presence of floodplains non-connected to the river (e.g., Frappart et al., 2005; Heimhuber et al., 2017). Due to the small area of the non-connected lakes present in the McKenzie delta, they are detected in our approach based on the use of MODIS images at 500 m of spatial resolution, as mixture areas (except during the June flood event where almost all the delta is inundated and all the flooded areas are connected to the river). Only the floodplains connected to river are considered in this study".

RC 18: Figure 3: A scale bar would make it easier to get an idea of the size of the index images that are shown in each panel. A.R 18: We chose to display the geographical coordinates on each map to have both the scale and the localization. This is why, we decided not to add a scale bar.
RC 19: Figure 5: You might want to keep the direction of your colour bar consistent (i.e. more days is red, less days is blue). A.R 19: We chose this color code to make the correspondence between blue and wetter conditions and red drier conditions. If this represents a big issue, we can modify the color bar.

RC 20: page 13 line 6 & Figure 6: Given that the surface water extent of the fi̧rst time step of your annual time period is always the highest, maybe you should have started the annual time period 1 month earlier, to ensure that you always capture the peak of the flood extent. I fi̧nd it problematic that you state that surface water extent is maximum in June, given that you never looked at May. A.R 20: We did not use the images from May as in most of the cases, there are still snow and ice. We added the following sentence in the manuscript: "Images from May were not used due to the presence of remaining snow and ice in the Mackenzie delta"

RC 21 : Figure 7: How about overlaying the classifi̧ed 500m pixel MODIS and Landsat image to highlight the differences. Give pixels where both agree one colour and then another colour for only water on Landsat and one for only water on MODIS. A.R 21: Following your comment, I've done this figure and introduced it in the supplementary script (Figure S4). Yellow pixels are corresponding to water in the both images (MODIS and OLI 500), light blue to water only for OLI 500 and dark blue to water only for MODIS.

Technical Corrections:

RC 22 : page 1, line 20: In this study, the dynamics of surface water extent and volume "are or were" analyzed from 2000 to 2015 by combining multi-satellite information from MODIS multispectral images at 500 m spatial resolution and river stages derived from ERS-2 (1995-2003), ENVISAT (2002-2010) and SARAL (since 2013) altimetry data. A.R 22: Corrected.

RC 23: page 2 Line 5: Discharge instead of discharger A.R 23: Corrected.
RC 24: page 7 line 10: you mean Hereafter? A.R 24: Âń Hereafter Âż and Âń There-after Âż, have close meanings. We chose Âń Thereafter Âż meaning Âń from then Âż.

RC 25: page 13 line 11: "." missing after MODIS? A.R 25: Corrected.

RC 26: page 13 line 20: mistake in "Frthat" A.R 26: Corrected.

RC 27: page 13 line 23: Sentence starting with "Besides, these..." is gramatically wrong and I do not get the point. A.R 27: This sentence was modified as follows: "These products provide a unique long-term dataset that allows a continuous monitoring of the changes affecting the surface water reservoir before the launch of the NASA-CNES Surface Water and Ocean Topography (SWOT) mission in 2021. The recent launches of Sentinel-1, Sentinel-2 and 3 offer new opportunities for flood monitoring at higher spatial ($\sim$10 m) and temporal (a few days) resolutions".

Please also note the supplement to this comment:
https://www.hydrol-earth-syst-sci-discuss.net/hess-2017-170/hess-2017-170-AC2-supplement.pdf

**Supplement:**

*Supplement of information*

**Quantification of surface water volume changes in the Mackenzie Delta using satellite multi-mission data**

Cassandra Normandin[1], Frédéric Frappart[2, 3], Bertrand Lubac[1], Simon Bélanger[4], Vincent Marieu[1], Fabien Blarel[3], Arthur Robinet[1] and Léa Guiastrennec-Faugas[1]

[1] EPOC, UMR 5805, Université de Bordeaux, Allée Geoffroy Saint-Hilaire, 33615 Pessac, France

[2] GET-GRGS, UMR 5563, CNRS/IRD/UPS, Observatoire Midi-Pyrénées, 31400 Toulouse, France

[3] LEGOS-GRGS, UMR 5566, CNRS/IRD/UPS, Observatoire Midi-Pyrénées, 31400 Toulouse, France

[4] Dép. Biologie, Chimie et Géographie, groupe BOREAS and Québec-Océan, Université du Québec à Rimouski, 300 allée des ursulines, Rimouski, Qc, G5L 3A1, Canada

*Correspondence to*: Cassandra Normandin (cassandra.normandin@u-bordeaux.fr)

**Figure S1: Index map and associated histogram in July 2013 for (a) EVI for MODIS, (b) EVI for OLI 500, (c) LSWI for MODIS and (d) LSWI for OLI 500**

[Figure]

**Figure S2: Map of difference between EVI and LSWI indices and associated histogram in July 2013 for (a) MODIS and (b) OLI 500**

[Figure]

**Figure S3: Surface water extent estimated in July 2013 using (a) MODIS at 500 m (b) OLI at 500 m and (c) OLI at 30 m. Permanent water bodies appear in blue and inundated areas in light blue.**

[Figure]

**Figure S4: Comparisons of surface water areas detected using MODIS at 500 m and Landsat-8 images resampled at 500 m of spatial resolution for Landsat-8 images acquired on 04/07/2013 (a) and 05/08/2013 (b). Surface water only detected using MODIS, Landsat-8 resampled and detected by both appears in blue, light blue and yellow respectively.**

[Figure]